# Hydrothermal Alteration at the San Vito Area of the Campi Flegrei Geothermal System in Italy: Mineral Review and Geochemical Modeling

Monica Piochi [1,*], Barbara Cantucci [2,*], Giordano Montegrossi [3,4] and Gilda Currenti [5]

1. Istituto Nazionale di Geofisica e Vulcanologia, Osservatorio Vesuviano, 80124 Naples, Italy
2. Istituto Nazionale di Geofisica e Vulcanologia, Sezione di Sismologia e Tettonofisica, 00143 Rome, Italy
3. Centro Nazionale delle Ricerche, Istituto di Geoscienze e Georisorse, 50121 Florence, Italy; giordano.montegrossi@igg.cnr.it
4. Consorzio Interuniversitario Nazionale per la Scienza e Tecnologia dei Materiali, 50121 Florence, Italy
5. Istituto Nazionale di Geofisica e Vulcanologia, Osservatorio Etneo, 95125 Catania, Italy; gilda.currenti@ingv.it
* Correspondence: monica.piochi@ingv.it (M.P.); barbara.cantucci@ingv.it (B.C.)

**Abstract:** The Campi Flegrei geothermal system sets in one of the most famous and hazardous volcanic caldera in the world. The geothermal dynamics is suspected to have a crucial role in the monitored unrest phases and in the eruption triggering as well. Numerical models in the literature do not properly consider the geochemical effects of fluid-rock interaction into the hydrothermal circulation and this gap limits the wholly understanding of the dynamics. This paper focuses on fluid-rock interaction effects at the Campi Flegrei and presents relevant information requested for reactive transport simulations. In particular, we provide: (1) an extensive review of available data and new petrographic analyses of the San Vito cores rearranged in a conceptual model useful to define representative geochemical and petrophysical parameters of rock formations suitable for numerical simulations and (2) the implemented thermodynamic and kinetic data set calibrated for the San Vito 1 well area, central in the geothermal reservoir. A preliminary 0D-geochemical model, performed with a different contribution of $CO_2$ at high (165 °C) and low (85 °C) temperatures, firstly allows reproducing the hydrothermal reactions over time of the Campanian Ignimbrite formation, the most important deposits in the case study area.

**Keywords:** geothermal system; hydrothermal alteration; Campi Flegrei; San Vito 1 well; reactive model





## 1. Introduction

Geothermal systems are common at active volcanic calderas (e.g., [1–5]). They result by the advective and convective cooling processes favored in the long volcanic quiescence and weak tectonic settings.

The Campi Flegrei volcanic-geothermal system (Italy; Figure 1) sets in one of the most famous and hazardous caldera in the world (e.g., [6–10]). This site was extensively investigated (e.g., [5,9,11–26]) with a long lasted exploration interest since Greek-medieval periods (e.g., [27]) up to the present [6,7,10,28–30]. Hundreds of boreholes (e.g., [5]) have drilled all over the volcano subsurface; a few by the Azienda Geologica Italiana Petroli (AGIP) and the Società Anonima Forze Endogene Napoletane (SAFEN) reached depths of 1000–3000 m providing exceptional physicochemical information and rock cores of the deeper reservoir at two main areas, namely Mofete and San Vito [6–8,11].

The volcanic-geothermal field coincides with an 8 km-wide basin generated by the caldera event at 39 ka (the Campanian Ignimbrite eruption) and is mostly infilled by the Campanian Ignimbrite and later erupted pyroclastic deposits (e.g., [5,10,31] and references therein). It shows a temperature up to 350 °C measured at −3000 m, geothermal gradients in the range 100–250 °C km$^{-1}$, discrete aquifers [6,7,32–34], acid brine-types

and saline fluids [8,28], hydrothermal discharges (hot springs, wells and fumaroles) and a huge release of $CO_2$ (ca. 1500–4000 td$^{-1}$) (e.g., [9]). However, hydrothermal circulation involves dominantly meteoric waters (and seawater) with minor magmatic contributions (i.e., juvenile $H_2O$, $CO_2$, $H_2S$ and $NH_4$) (e.g., [14,26,35]). Hydrothermal alteration is extensive. The deeper reservoir rocks (ca. $< -2500$ m) are modified by thermometamorphism (e.g., [5,6,8,31]). Upwards, propylitic alteration gradually transforms into the argillitic facies of the impermeable shallow covering and proceed into the acid sulfate alteration at the discharge areas [6–8,11,23,36,37].

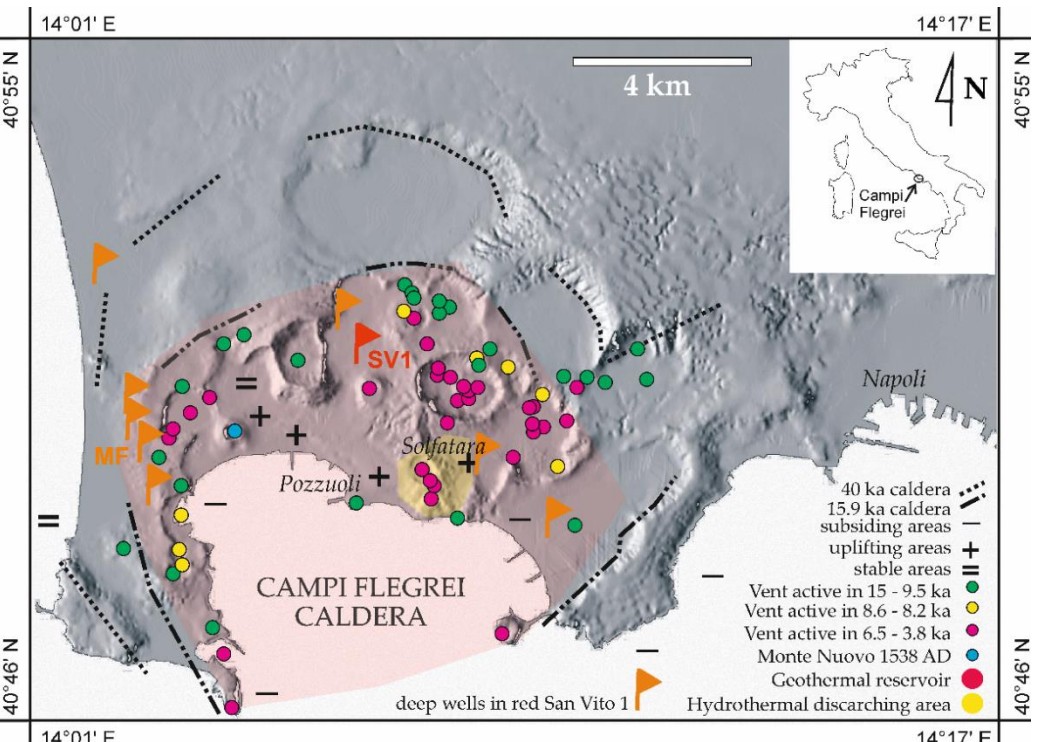

**Figure 1.** The Campi Flegrei volcano with the main information about physical aspects, ages, geological relations, relief, vents active in the most recent volcanic history, deformation areas, hydrothermal and discharge zones, SAFEN-AGIP's deep wells (modified from [5]). Coordinates in degrees–minutes–seconds. In red is the San Vito 1 well (SV1) object of this study. MF is the Mofete wells area. DTM (digital terrain model) of the area from the Laboratorio di Geomatica e Cartografia of the Istituto Nazionale di Geofisica e Vulcanologia, Osservatorio Vesuviano, Italy.

The field has been the site of tens of monogenic eruptions until 1538 AD (e.g., [21,22,38]). Notably, its central portion inflates during ongoing periods of intense seismicity and hydrothermal anomalies (e.g., [13,25,26,39,40]).

The role of the geothermal system in the monitored unrest phases leaving meters of ground uplift is suspected and a strong debate exists on the real dynamics (e.g., [13,14,22,26,41–43]). Moreover, the involvement of the geothermal system in phreatomagmatic explosions that has dominated the volcanic history of the Campi Flegrei is expected. Several studies modeled hydrothermal circulation by using the TOUGH2 code [44] with the aim to explain ground deformation, fluid geochemistry and gravity data on the basis of subsurface information at the Mofete wells (e.g., [45–50]). The 2D, 3D, symmetric and no-symmetric models were all able to simulate data by the injection of hot fluids in an either fractured or homogeneously or heterogeneously permeable reservoir, from a source at different depths (−1500 m and −3000 m). Petrillo [49] highlighted the effects of the heterogeneous distribution of the rocks' physical properties in the hydrothermal circulation.

However, the actual models do not consider the geochemical effects of the fluid-rock interaction (and associated variation of petrophysical properties) on the hydrothermal

circulation. In the frame of a longer-term project, aimed to perform reactive transport simulations at Campi Flegrei, the objective of this study is to collect mineralogical information and calibrate the thermodynamic dataset with actual data to be used as input for numerical models. The area drilled by the San Vito 1 well (Figure 1) was selected as a case study.

The paper is structured as follows: after describing strategy and materials (Section 2), Section 3 provides the downhole lithological and mineralogical data at the study site achieved owing to an extensive review of the original unpublished documents by AGIP and SAFEN (dated at 1982) and internal proceedings [6]; Section 4 furnishes the new petrographic and mineralogical results obtained on cores by optical and energy dispersive system-back scattered electron microscopy describing the hydrothermal alteration; Section 5 rearranges the literature and newly acquired data and provides initial reservoir stratigraphy and mineralogy as input data for the transport reactive modeling. The final two sections are dedicated to describe and discuss modeling results that focused on the most widespread Campanian Ignimbrite deposits because it represents an important formation fed by deeper sourced $CO_2$ fluids.

## 2. Site Study, Strategy, Materials and Methods

The San Vito 1 well (Figure 1) reached depths of over 3000 m but the collected materials received limited attention, probably because technical problems seemed to favor the other drillings in Mofete or Mofete was a more promising exploring area [6]. Though, the San Vito area is of particular interest for several reasons: (1) its central position in the field; (2) the highest values (up to 400 °C) and time stability of the subsurface temperature [8]; (3) closeness to the discharge areas, i.e., the Solfatara tuff cone, the only one touched by powerful hydrothermal activities with hot fumaroles, thermal springs, mud pools, diffuse outgassing and solfataric alteration (e.g., [23]); (4) location at the limit of the portion subject to the maximum ground uplift (in excess of 3 m with 5 cm/year to 1.5 cm/month) and seismicity (more than 16,000 low-magnitude earthquakes) (e.g., [13,25,51]); lastly, (5) the concentration of most vents erupting in the past 5 ka (e.g., [21]).

We use multiple review and analytical approaches to provide a simplified conceptual picture of the medium reservoir and the related dataset of rock lithology, mineralogy, rock and fluid chemistry, permeability and porosity useful for numerical simulations.

Cores, unpublished reports (AGIP-SAFEN, 1982) and internal proceedings by [6] are the essential basis for reservoir constraints and conceptualization in this study. Therefore, we started with the review of original unpublished documents by AGIP and SAFEN (dated at 1982) and those internal proceedings [6]; several authors must be thanked for the lot of work done that is not easily available in the online literature resources (Appendix A). The review allowed the description of the authentic lithology at the San Vito 1 well and furnished the original information of intercepted rocks and mineral data of the downhole, reported in Section 3. Both achievements that are lacking in the available literature are crucial for studies aimed to gain an improved picture of the geothermal reservoir. To note, rock mineralogy, determined during drilling from the on-site petrographic analysis of cuttings and cores, occur in the AGIP's graph papers. Type and abundance of minerals that AGIP indicated to be determined by X-ray diffraction and electron microprobe analyses during drilling were manually recovered from the above cited graph papers.

To check and well-define the original lithology we inspected cored rocks at both macroscopic and microscopic scales and the results of our inspection are in Section 3.1.1. Then, we selected and focused more detailed textural and mineralogical investigations on those cores allowing us to gain further details for the reservoir reconstruction from the surface down to −1800 m (Section 4). We considered this depth range where the rock features and mineral alteration give the possibility to unravel the protolith formation units and primary phases with respect to secondary mineralization (Figure 2 and Sections 3.1.2, 4 and 5.1). Additionally, the distance from the deeper (≥4 km, [5,17]) magmatic intrusion(s) permits the use of a unique source of heat and fluids. Moreover, focusing on shallower deposits limits the needed constraint and simplifies the numerical domain in the prospective of

future reactive transport simulations. Notably, due to the absence of lava rocks cored at San Vito 1, we considered lavas that occurred at the Mofete 1 well.

Rock texture and mineral composition were determined on the selected cores prepared as thin sections coated by rod graphite in the ZEISS electron microscope (EDS-BSEM) after preliminarily observations under the plane-polarized, cross-polarized and reflected light of the optical microscope (Axiop 40 by ZEISS). The ZEISS instrument is a SIGMA field emission scanning electron microscope, equipped with an XMAN micro-analysis system by Oxford, controlled by SMARTSEM 5.09 (Carl Zeiss, Oberkochen, DE, Germany) and AZTEC 3.0 (Oxford Instruments, High Wycombe, UK, United Kingdom) softwares (Osservatorio Vesuviano, Istituto Nazionale di Geofisica e Vulcanologia, Naples, Italy). Operating conditions were 15 kV accelerating voltage, 50–100 mA filament current, 8.5 working distance, 5–10 nm spot size and variable acquisition time (several to tens of seconds).

Building on previous studies on both surface and subsurface deposits ([5,7,8,11,29,52] and references therein) and AGIP-SAFEN achievements, the new analytical results allow constraining protolith (primary in the model) and hydrothermal (secondary) phases and the investigation of the hydrothermal alteration as well.

Data have been further rearranged to define the conceptual model concerning the initial reservoir stratigraphy and mineralogy that can be used by other authors.

To provide a first evaluation of reactive processes at the Campi Flegrei, we developed a geochemical model of the Campanian Ignimbrite (ca. 1000–1500 m of depth), which is the most important widespread deposits of the area (see Section 5.1) and represents the most surficial reservoir fed by fluids of magmatic origin, mainly $CO_2$. The numerical model setup, including flow and geochemical conditions, are described in Section 5.2 together with the dataset we used after rearranging new and reviewed data. The TOUGHREACT [53,54] commercial code integrated with the EOS2 module [55,56] was employed for batch-reaction simulations to allow input consistency with further reactive transport simulations. This model properly describes fluids in $CO_2$-rich reservoirs ($0\,^{\circ}\text{C} \leq \text{T} \geq 350\,^{\circ}\text{C}$), accounting for the non-ideal behavior of gaseous $CO_2$ and dissolution of $CO_2$ in the aqueous phase with the heat of solution effects. The 0D geochemical simulations were performed at different $PCO_2$ conditions and temperatures (i.e., 165 and 85 $^{\circ}$C) compatible with the investigated depth range (see Section 5.2.1). Temporal mineral alteration was simulated, subjected to a variable $CO_2$ contribution, at high and relatively low temperatures. In each simulation, $PCO_2$ and temperature were maintained constant over time. Simulations were run for thousands of years until a near kinetic equilibrium among the mineral was reached.

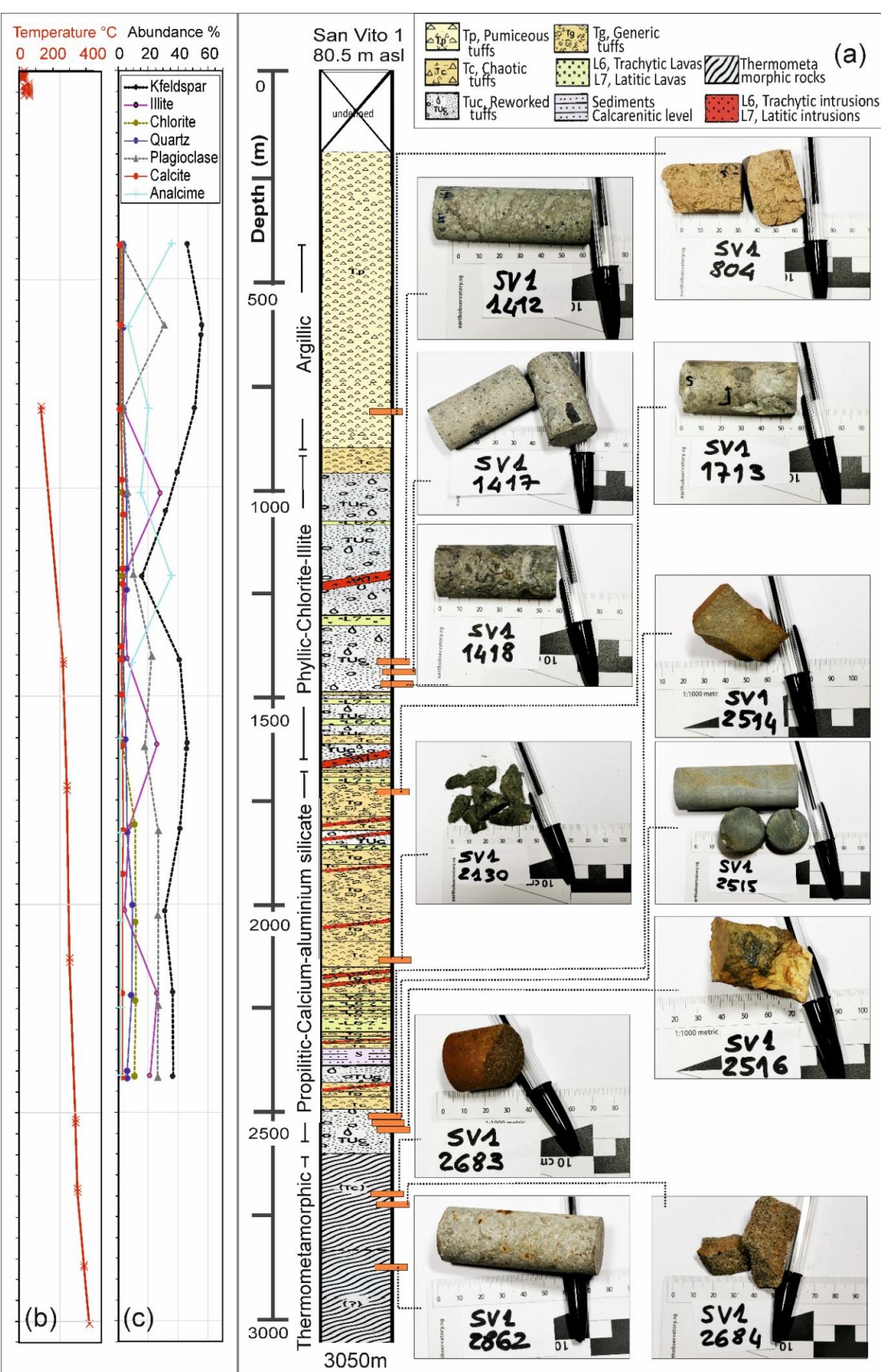

**Figure 2. (a)** Downhole San Vito 1 stratigraphy with the depth location and photo of cores, alteration facies and distribution of **(b)** temperature and **(c)** mineral abundance. Data source: Piochi et al. [5] for the lithology, De Vivo et al. [8] for the temperature,

AGIP [6] for the alteration facies (see also Appendix B) and AGIP's (1982) internal documents for the mineral abundance. The scale in photos is in millimeters. An 804 m-core: zeolitized yellowish to reddish tuff made of a chloritized trachytic vacuolar glassy matrix with millimeter-sized pumice, scoria and lithic clasts having a lati-trachytic nature. A 1412 m-core: chaotic tuff containing millimeter-sized pumice, scoria and lithic clasts. A 1417 m-core: chaotic tuff with millimeter-sized pumice, scoria and lithic clasts. A 1418 m-core: vacuolar homogeneous rocks, likely a protolithic ashy-to-sandy tuff. A 1713 m-core: tuff, ashy-to-sandy with low-to-highly crystalline scoria; it includes vacuolar trachytic lava clasts with feldspar acicular microcrysts. A 2130 m-core: chaotic tuff with rounded trachytic lithic clasts. The 2514-2515-2516 m-cores: gray tuff with a sandy massive matrix having a trachy-latitic nature. A 2683 m-core: homogeneous porphyritic lava. A 2684 m-core: massive fine grained rock of a subvolcanic nature. A 2862 m-core: homogeneous rock; recrystallized. The Campanian Ignimbrite formation object of geochemical modeling is in the 1000–1800 m depth range. Refer to the text for details and Appendice B and Appendice C for downhole alteration facies and for lithological features and minerals occurrence of cores.

## 3. The San Vito 1 Well

### 3.1. Reviewed Subsurface Data

3.1.1. Downhole Lithology

Table S1 lists the salient features of rocks intercepted through the San Vito 1 well from earlier descriptions, together with core data on density, porosity and permeability. The authentic lithography is in Figure 2 with alteration facies (Appendix B). This one matches descriptions by Bruni et al. (in [6]) and the simplified representation in [31]. It was made of preponderant pyroclastic deposits (i.e., we calculated ca. 705 m of pumiceous rocks, ca. 345 m of chaotic tuffs, ca. 395 m of generic tuffs and ca. 1128 m of tuffites) down to the thermometamorphic rocks at depth >ca. 2600 m, but also includes minor lithologies omitted in other reconstructions, i.e.,:

- A ca. 15 m-thick calcarenitic level at −2378 m within a succession of conglomerate layers with volcanic elements and rare silty-marly and mycritic sedimentary clasts;
- Hundreds of meters of lavas at depths deeper than ca. 1000 m;
- n.10 (less than tens meters each) intrusive or subvolcanic bodies with the alkali-trachytic to latitic nature between −1200 and −2450 m.

Building on Bruni et al. (in [6]) and our inspection of cores (further details in Appendix C), the authentic lithography at the San Vito 1 well includes:

- In the depth range 200–905 m: incoherent (between −200 and −330 m) to lithified (between −420 and −905 m) pyroclastic deposits, bearing trachy-latitic pumices, scoria (mostly between −200 and −400 m) and scarce lava clasts. The core at −804 m shows a zeolitized yellow tuff (Figure 2a), somewhat argillified and chloritized (see next Sections 3.1.2 and 4.1, and Appendix B). It is massive, friable, heterogeneous both in the type of components and grain sizes and significantly vacuolar, showing the usual aspect of some outcropping tuffs in the area. Denser and more porous clasts and crystals (mostly feldspars and minor pyroxenes and biotites) are distinguishable at the macroscopic inspection within a sandy-like matrix.
- In the depth range 905–975 m: chaotic tuffs. Bruni et al. (in [6]) indicated "re-arranged pyroclastic products, rich in phenocrysts (sanidine, plagioclase, aegirinaugite, biotite, detrital quartz and trachytic-to-latitic lava clasts), and presence of fragments of fossils and glauconite".
- In the depth range 975–1660 m: chaotic fossiliferous alternating with tuffites, subordinate trachytic-to-latitic lavas and rare chaotic tuffs. At ca. 1200 m the first latitic subintrusive body appears. Figure 2a shows the chaotic tuff samples at ca. 1412–1418 m of depth. The cores are very similar to each other's, being greenish in color, compact, vacuolar and heterogeneous due to the occurrence of lava- and pumice-type clasts with a size up to several millimeters in an ashy-to-sandy matrix. Bruni et al. suggested "a coastal depositional environment (littoral fauna association)". The facies here varies from argillic to phyllic (see next Sections 3.1.2 and 4.1 and Appendix B).
- In the depth range 1660–2365 m: an alternation of subaerial chaotic tuffs, latitic tuffs, trachytic-to-latitic lavas and subvolcanic latitic and trachytic bodies. The sample from the depth of 1713 m is shown in Figure 2a; it was cored at the deeper limit of the

phyllic facies (see next Sections 3.1.2 and 4.1 and Appendix B). It has a similar aspect to the three shallower cores, i.e., greenish, bearing clasts and crystals (feldspars and pyroxene), although it appears more friable and has a higher porosity. The friability is higher in the deeper core at 2130 m of depth, which has a tendency to be more homogenous and shows a stronger greenish color.

- In the depth range 2365–2500 m: an alternation of chaotic tuffs, chaotic tuffites, tuffs, subvolcanic trachy-latitic bodies and the calc-arenitic level.
- In the depth range 2500–2600 m: fine grained chaotic tuffites with trachy-latitic features. The cores display the variable aspect of the deposits (Figure 2a). The cores at 2514 m and 2516 m of depth are brownish to reddish, granular, heterogeneous and cohesive. They strongly differ from the 2515 m-core that is a compact greenish homogeneous block. At these depths, the hydrothermal alteration is stronger with silicification and propylitization processes that obliterated the original rock (see Section 3.1.2 and Appendix B).
- In the depth range 2600–2840 m: conglomerates with trachy-latitic volcanic clasts and sedimentary elements. Bruni et al. (in [6]) indicated "(silty-marly and mycritic)" features for sediments, in agreement with the core look (Figure 2a). These cores at 2683 m and 2684 m have a rust-like appearance and granular texture. The shallow one is more compact. Here, intense thermometamorphism has been described with scapolite and neogenic biotite and actinolite (see Appendix B).
- In the depth range 2840–3046 m: metatuffs or thermometamorphic rocks with recrystallization and obliteration of the original rocks (see Appendix B). Bruni et al. (in [6]) suggested a possible "pyroclastic origin (there still have traces of pumices and lavic clasts". The macroscopic aspect of the metatuff is in Figure 2a. It is a massive greenish homogeneous block with a reddish portion.

By considering the downhole location of the lava units, the intrusions, the transition between the phyllic and the propylitic facies and the sediments, the depth around 1500–1700 m represents an important subsurface discontinuity of the geothermal system.

### 3.1.2. Downhole Mineralogy

Table S1 reports semiquantitative data on mineral phases measured during AGIP's drilling activity and this has never been published before. The manually recovered relative abundance was plotted along the reconstructed stratigraphy in Figure 2a. Detected phases by AGIP were calcite, illite, chlorite, quartz and predominantly analcyme/zeolite and feldspars. Calcite has low abundance (<5%) and disappears at ca. 2500 m, as already reported [5]. At shallow depths, chlorite and quartz were at ca. 3% and 5%, respectively. Quartz reached up to 10% between ca. 1310 and 1500 m and from ca. 1900 m downhole; chlorite progressively increased from 1800 to 1900 m where it was at ca. 10%, a value that remained in the rest of the borehole. The illite was generally at a few percent showing a single spike of ca. 25% in the chaotic tuffs at around 1100 m, being at a slightly higher value of 5% between ca. 1860 and 2100 m of depth and increasing up to 25% in the portions deeper than 2300 m. Everywhere, alkali-feldspar and plagioclase were the most abundant phases up to 40% and 25%, respectively. Analcyme/zeolite phase was abundant (ca. 20–30%) down to −1700 m and then decreased at less than a few percent.

We revealed that the variation of the XRD-derived mineral abundances did not have a direct relation with the lithology and log parameters (Table S1). Nevertheless, at ca. −1700–1900 m the quartz started to increase (ca. 7%–10%), whereas the analcyme/zeolite dropped down (ca. 10% to <2%) and the porosity appeared to decrease (roughly ca. 35%–25%). At this level there was the transition between the phyllic and the propylitic facies.

## 4. Analytical Results

### 4.1. Protolith and Hydrothermal Alteration Minerals

Representative results of new optical and EDS-BSEM investigations on rocks cored at depths shallower than 1800 m, the object of this study (hereafter rocks reservoir), are

presented in Figures 3–6 and in the Supplementary Material (Figures S1 and various panels in S2), top downward in the hole.

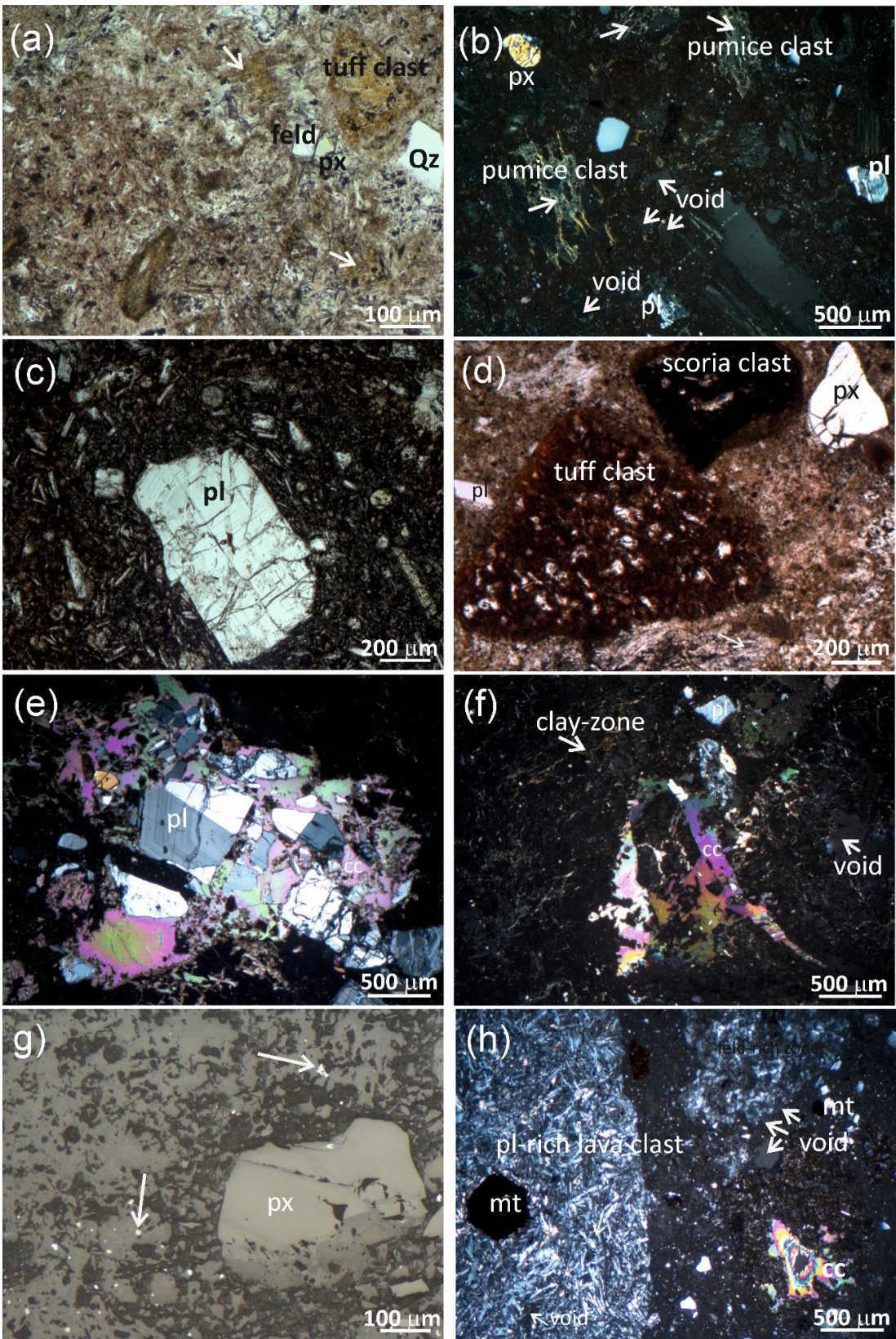

**Figure 3.** The reservoir rocks cored at San Vito 1 well under the optical microscope: (**a**–**c**) the yellow tuff cored at 804 m of depth from the argillic facies and (**d**–**h**) the chaotic tuffs (representing the modeled Campanian Ignimbrite formation) cored at depths

of 1412 and 1417 m from the phyllic facies (refer to Figure 2a for the downhole location and aspect of cores). The light is plane-polarized in (**a**,**d**), cross-polarized in (**b**,**c**,**e**,**f**,**h**) and reflected in (**g**). (**a**) The tuff cored at 804 m consists of a yellowish, nearly homogeneous and low vacuolar matrix with various tuff clasts of less than 1 mm. (**b**) The tuff cored at 804 m has a fully extinct matrix under the cross-polarized light, due to the analcyme (and possibly minor glass) phases (see Figures 5 and 6 and the main text) and the voids as well; the included pumice relicts emerge from the extinct matrix due to the strong alteration. The 804 m tuff includes few phenocrysts, of mostly a volcanic (feldspars, plagioclase and pyroxene) and hydrothermal (quartz) nature (**a**,**b**). (**c**) The tuff cored at 804 m includes fresh lava clasts, the matrix of which is generally rich of feldspar microcrysts and may include larger euhedral zoned plagioclase phenocrysts. (**d**) The chaotic tuffs from 1412 m of depth shows a reddish and low vacuolar matrix including variable crystalline and vesicular scoria-like clasts; it contains pyroxene grains. (**e**,**f**) The chaotic tuffs show a matrix that is partially extinct and microcrystallized (due to analcyme and albite, respectively, as described in Figures 5 and 6 and the main text) under the cross-polarized light. It is porphyritic; the image (**e**) displays a plagioclase clot disrupted and partially replaced by calcite, except for the larger plagioclase phenocryst. The image (**f**) shows that the tuff characterizes for void fractures filled by calcite. (**g**) The pyrite can be rarely found in lava clasts (the clast is the same of that in Figure S1a under the cross polarized light). (**h**) The lava clasts included in the tuff have high abundance of plagioclase microlites and occasional magnetite; the image well shows the different crystallization of the tuff matrix and the lava clasts in the 1417 m core. Arrows indicate (**a**) highly zeolitized, (**b**,**f**) clay-rich and (**d**) feldspar-rich portions, (**b**) pumices; (**g**) pyrite grains or (**b**,**f**,**h**) voids (light gray areas under the cross-plane light). Abbreviations: feld, alkali-feldspar; px, pyroxene; pl, plagioclase; Qz, quartz; cc, calcite; mt, magnetite. Refer to the text for additional details.

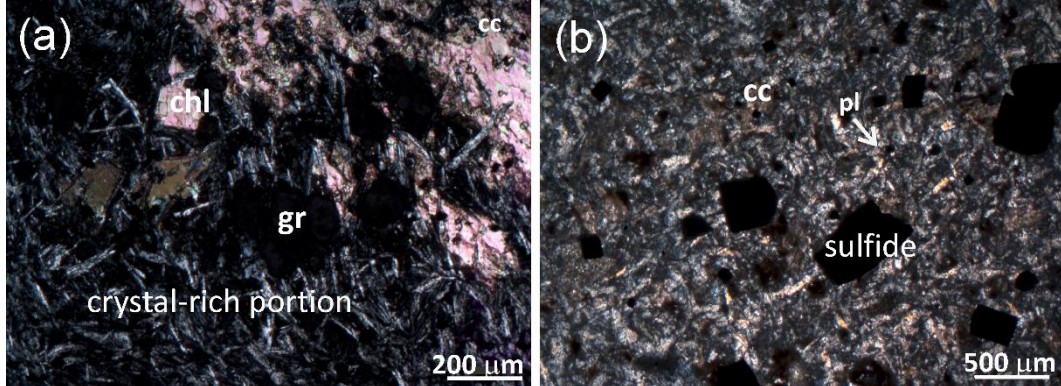

**Figure 4.** The cored reservoir rocks under the cross-polarized light of the optical microscope: (**a**) the chaotic tuff (representing the modeled Campanian Ignimbrite formation) cored at 1713 m of depth from the propylitic calcium–aluminum–silicate facies in the San Vito 1 borehole (refer to Figure 2a for the downhole location and aspect of the core) and (**b**) the lavas cored at depths of 1500 m from the phyllic facies in the Mofete 1 borehole. (**a**) The San Vito 1 chaotic tuff consists of a low vacuolar and locally crystalline matrix; it presents widespread calcite and chlorite set in the intergrains and shows secondary garnet crystals. (**b**) The Mofete 1 lavas show a highly crystalline matrix with several percent of sulfides and void-filling calcite; the plagioclase microcrysts are altered. Abbreviations: pyroxene; pl, plagioclase; chl, chlorite; cc: calcite; gr: garnet. The plane-polarized image of the San Vito 1 and Mofete 1 cores are in Figure S1. Refer to the text for additional details.

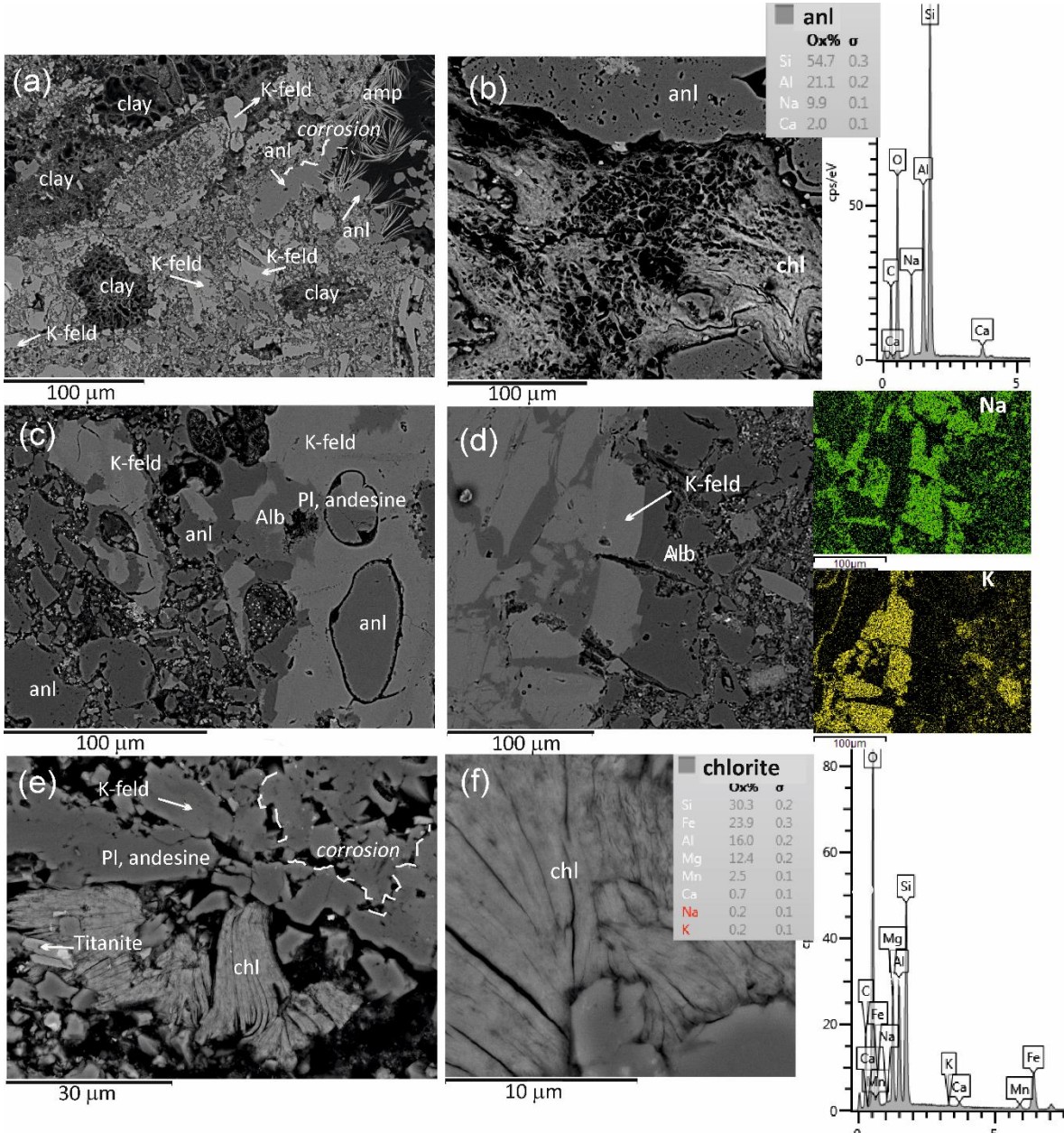

**Figure 5.** The reservoir rocks cored in the San Vito 1 borehole under the electron microscope (**a**–**f**) with representative EDS spectra and maps (on the right). (**a**,**b**) The yellow tuff cored at 804 m of depth from the argillic facies, shows a complex texture characterized by widespread K-feldspar microcysts constituting the matrix and clay filling voids. Analcyme with an anhedral habit and corroded edges is present as alteration products; it has a low Na content. EDS indicates chlorite compositions among argillic phases. (**c**,**d**) The chaotic tuff core from 1412 m of depth from the phyllic facies presents a mosaic of analcyme and K-feldspar microcrysts. Locally, K-feldspar is replaced by albite (see EDS maps for Na and K elements). (**e**,**f**) The chaotic tuff cored at 1713 m of depth from the propylitic calcium–aluminum–silicate facies characterized by the corroded K-feldspars and well-formed chlorites. Chlorite is Fe-rich (see the EDS spectra at the right bottom panel). Abbreviation: K-feld, K-feldspar; chl, chlorite; anl, analcyme; clay, argillic zones; Pl, plagioclase; Alb, albite, Amp, amphibole. Dashed lines indicate the main corrosion zone along the crystal rims.

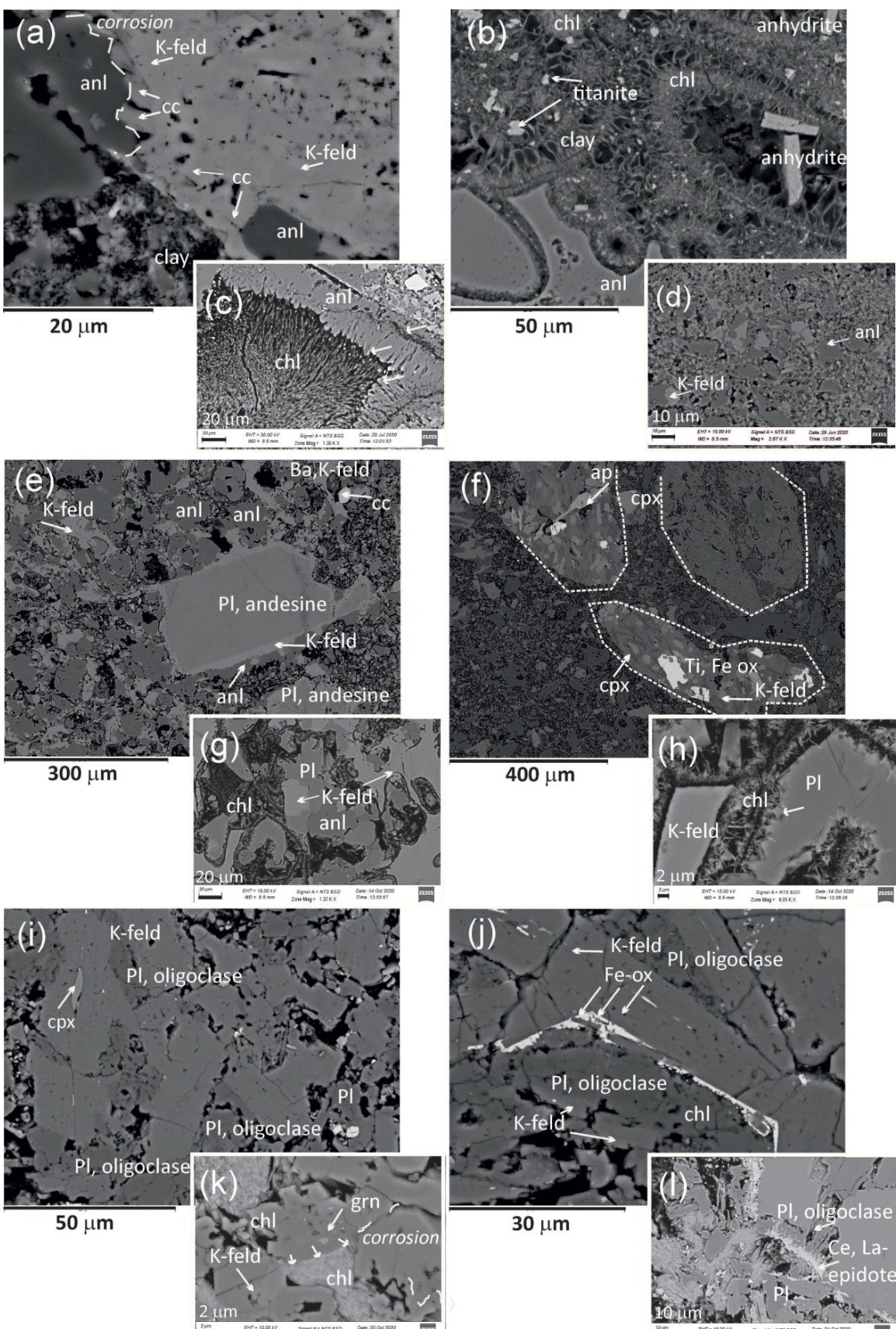

**Figure 6.** The reservoir rocks cored at the San Vito 1 borehole under the electron microscope. (**a**–**d**) The yellow tuff cored at 804 m of depth from the argillic facies, (**e**–**h**) the chaotic tuffs cored at depths of 1412 m from the phyllic facies and (**h**,**i**) the chaotic

tuff cored at 1713 m of depth from the propylitic calcium–aluminum–silicate facies (refer to Figure 2a for the downhole location and aspect of cores). (**a**,**b**,**e**,**g**) The analcyme forms a mosaic of alteration on the protolith K-feldspars. (**b**) The clay phases develop as a complex network among the K-feldspars and the analcymes. (**c**) The clay develops on analcyme. (**e**,**g**,**h**) The rocks at 1412 m of depth contain a higher abundance of plagioclase; the plagioclase is rimmed by K-feldspar. (**f**) The lithic grains are generally richer in plagioclases and pyroxenes with respect to the host rocks; sometimes they are nearly holocrystalline. (**i**) The protolith plagioclase and K-feldspars in the core from 1713 m of depth are strongly corroded leaving irregular space voids. (**j**) The narrow spaces between feldspars in the 1713 m core are filled by secondary Fe-oxides. (**k**) The chlorite fills the voids of the 1713 m cored rock. (**l**) The 1713 m core shows numerous minute epidotes, some are rich in La and Ce. Abbreviation: K-feld, K-feldspar; chl, chlorite; grn, garnet; anl, analcyme; clay, argillic zones; Fe-ox, iron oxides; cc, calcite; Pl, plagioclase; cpx, clinopyroxene.; ap, apatite. Dashed lines indicate the corrosion zone along the crystal rims; in (**f**) dashed lines encircle the lava grains.

The petrographic images show the typical textural features of tuffs for rocks cored at 804 (Figure 3a–c), 1412 (Figure 3d–g) and 1417 m depth (Figure 3h), with some phenocrysts and pumice, scoria and lava grains included in a finer matrix. The embedded pumices were generally more altered, locally showing elongated shapes and tube vesicles (Figure 3b). Scoria and lava fragments had more defined shapes, the scoria being smoother (Figure 3b,d,h). The lava clasts were more crystalline and porphyritic (Figures 3c,h and S1a). The 1713 m sample had a trachytic intersertal-type texture (Figures 4a and S1b). Intersertal texture and low porphyriticity were detected in the lavas drilled at Mofete 1 (Figures 4b and S1c) and we inferred that it characterizes the lavas (not cored) at San Vito 1. Under the optical microscope, alkali-feldspars and plagioclase were dominant on pyroxene, calcite, magnetite, apatite, quartz and garnet. Clay and analcyme/zeolite were not easy to be distinguished under the optical microscope; the first should be at the origin of the iridescent and greenish aspect of the shallower tuffs (Figure 3a,b,f).

The EDS-BSEM results allowed better appreciation of the different textural and chemical features of the subsurface rocks (Figure 5). The yellowish tuff from 804 m exhibited a very complicated matrix consisting of a network of argillic formation disseminated in voids and fractures and zones with feldspar and analcime that determined different porosities (Figure 5a,b). Clays had pore-filling and pore-bridging occurrences. The Campanian Ignimbrite appeared more compacted and was featured albite at 1412 m (Figure 5c,d) and chlorite at 1713 m (Figure 5e,f). Feldspar and analcime zones presented a very low abundance of calcite (Figure 6a). In the 804 m-core the analcime defines the limit of clay formations (Figure 6b); clays appeared forming on dissolution gulfs of analcime (Figure 6c). This later always formed a mosaic-like texture with the alkali-feldspar (Figure 6d,e). The 1412 m-core maintained xenolith clasts almost in the form of texturally isolated grains, sometimes with a high crystallinity (Figure 6f); the tuff matrix displays a complex transition between plagioclase, alkali-feldspar and analcime compositions at the micrometric scale (Figure 6g); in the pore space between these minerals, chlorite develops (Figure 6h). The crystals were packaged in the 1713 m-core, the micropores of which were determined by irregular spaces between adjacent crystals (Figure 6i,j).

The optical and EDS-BSEM investigations corroborated the mineral type and abundance recorded by AGIP, providing additional details on the alteration processes. In particular, the EDS-BSEM inspection indicates alkali feldspars and plagioclases as major phases with K-feldspars being the most abundant (Figures 5 and S2). Pyroxene is present, but it can be considered scarce, occurring mostly as phenocrysts (Figures 3b,g and S1a). Both feldspars mostly occurred in microcrysts and subordinately as phenocrysts (Figures 3c,e and 5c,d), though plagioclase was concentrated in the lithic lava clasts and lavas as microcrysts (Figures 3c,h, 4a,b, 6f and S1a). Most of the tuff matrix was made of the K-feldspar microcrysts (Figures 5a,c and 6d). EDS indicates a measurable abundance of Ba for some K-feldspar grains and shows an oligoclase to andesine composition for plagioclase (Figure S2). Albite was further detected in the only 1412 m sample where it determined irregular patches on feldspars (Figure 5c,d); it reflected local secondary albitization processes. The most common phases were clay and analcyme/zeolites. The glass phase is virtually absent

because of the alteration, although it was certainly present as the protolith phase due the lithological nature of rocks and the occurrence of low porphyritic pumice and scoria clasts (Figure 3b,d; see next Section 5).

At the EDS-BSEM, clays formed an open felted network of minute poorly crystallized plates in the SV1 804 core (Figure 5a,b) and occurred in well-crystallized grains in the SV1 1713 core (Figure 5e,f), filling pores and void fractures. Clays were mostly chlorites (see also Figure S2), although those interlayering with smectites in the core tuff from 804 m of depth would be object of further investigations. Chlorite, with a frequent rosette morphology (not shown), was pervasive in the sample SV1 1713. It was the reason of a high birefringence reaching violet tones under the cross-polarized light of the optical microscope (Figures 4a and S1b) and this feature matched with the Fe-richness measured at the EDS (Figures 5f and S2). The EDS-BSEM indicates that analcyme was highly abundant in the core from 804 m of depth and, although at the lowest extent, in the 1412 m sample, whereas it disappeared in the other studied samples. Analcyme/zeolite caused the wholly extinction of tuffs under the cross-polarized light (Figure 3b,f). The analcimization proceeded as a corrosion front at the expense of the K-feldspars (Figure 6a,e) or via perlitic-like structures (Figure 6b), making a mosaic of alteration (Figure 6d) and leaving several minute inclusions or pores (Figure 5b). The analcyme acts as substratum for nucleation of chlorite (Figure 6c). Pore-lining chlorite develops from the crystals towards the pore center forming a felted mal obstructing pores (Figures 5e and 6b,c,h).

In agreement with AGIP's XRD data, calcite and quartz were scarcely observed. The optical images show that calcite occasionally filled voids and void fractures of rocks cored at ca. 1412 and 1417 m (Figure 3e,f,h). Calcite was present in a clot of disrupted plagioclase (Figure 3e). Quartz can be sometimes observed in isolated grains (Figure 3a).

Other secondary phases were: titanite, epidote sometimes rich in La and Ce (Figure S2), garnet and Fe-oxides in the 1713 m sample (Figures 5e and 6b,j–l) and cubic pyrites (Figure S2) and well-formed crystals of anhydrite in pores and fracture-voids of the 804 m sample (Figure 6b). Notably, sulphides and calcites were rare in the studied San Vito 1 rocks (Figure 3g), whereas at Mofete it was noticed up to tens of percent by the area of galena and sphalerite (Figures 4c and S1b) and calcite is <10% [8,52]. Following the authors, the garnet at a depth higher than 1700 m is due to contact metamorphism with hot magma bodies. At 1713 m, the feldspars were corroded (Figure 6i); oxides filled the narrow spaces between adjacent feldspars (Figure 6j) and fine epidote crystals mantled the feldspars (Figure 6l).

As a whole, the optical and BSEM images suggest that the alteration manifests as analcimization, argillification, and, locally, albitization. It did not obliterate the original features of rocks. Most of the feldspar and pyroxene crystals appeared to nearly maintain their original size and aspect. Cored rocks have mineral assemblage and texture comparable to that of outcropping volcanic deposits, with the abundance of feldspars, minor pyroxenes and low porphyricity. Most of K-feldspars and plagioclases, i.e., phenocrysts and intersertal microcrysts, in studied cores are protolith crystals.

## 5. Numerical Modeling

### 5.1. Data Set for the Formation Reservoir

To define the dataset of geochemical model we only considered the succession shallower than 1800 m. As previously written (Section 2), the main reasons for this are the higher homogeneity of the succession in terms of the rock and mineral types at this depth range (Figure 2 and Sections 3.1.2 and 4.1) and the possibility to link the buried lithology with surface deposits (see below and Section 5.1.1). However, in this way we could exclude the rocks with very strong alteration processes and thermometamorphism that obliterate their primary features (see Appendix B). In addition, we were far from the magmatic intrusion(s) that can be considered as a unique source of heat and fluids and we eluded the "hardened" (La Torre and Nannini) [6] silicified-to-thermometamorphosed rocks that, likely, are permeable to fluids through fractures. On the other hand, for the shallower succession we were more confident to propose volcanological correlations with surface deposits.

Table 1 lists the mineral parameters of protolith deposits through the buried stratigraphy at the San Vito 1 well. We extrapolated the parameters by new and literature data on surface deposits and cores, on the basis of stratigraphic consideration and rearrangement of data, as explained below in Section 5.1.1; the needed input data for the geochemical model are shown for the selected Campanian Ignimbrite formation. Notably, since most outcropping tuffs are zeolitized, we showed evidence for the hypothesis that zeolites (not only analcyme) and smectite would be considered as primaries in the mineral setting. This possibility was not the object of this study and will be tested in future simulations.

**Table 1.** Mineral features of protoliths.

| 0–200 m, Incoherent Pumiceous Deposits, i.e., Pyroclastic Products < 14.9 ka | | |
|---|---|---|
| Features: phono-trachytic pumices, scoria and ashy-to-sandy deposits | | |
| | | Variability |
| Porphyritic with matrix (%) | | 75–68 |
| **Primary Minerals:** | **Size (μm)/Type** | **wt %** |
| Pl (labradorite or bytownite)—f | 400 large:500 long | 5 |
| K-Feld—f | 700 large:1000 long | 15–20 |
| K-Feld—m | 5 large:10 long | 5–17 |
| CPX (diopside)—f | 400 large:700 long | 3–5 |
| BT—f | 400 large:700 long | 2 |
| Glass | | 70–20 |
| ZE (phillipsite>chabazite)—m | 2 large:10 long | 0–40 |
| Anl—m | 50 diameter | 0–3 |
| CPX (diopside)—m | 40 large:70 long | 0–1 |
| **200–1000 m, Zeolitized Yellow Tuff—Probable Formation: Neapolitan Yellow Tuff** | | |
| Features: trachytic ashy-to-sandy tuff with minor pumiceous levels | | |
| | | Variability |
| Porphyritic with matrix (%) | | 90 |
| **Primary Minerals:** | **Size (μm)/Type** | **wt %** |
| SM—m | 1 large: 10 long | 0–6 |
| Pl (labradorite or bytownite)—f | 400 large:600 long | 2 |
| K-Feld—f | 600 large:1000 long | 5 |
| K-Feld—m | 80 large: 100 long | 15–25 |
| CPX (diopside)—m | 80 large: 100 long | 3 |
| BT—f | 100 large:300 long | 3 |
| Glass | | 69–23 |
| ZE (phillipsite>chabazite)—m | 1 large:5 long | 0–30 |
| Anl—m | 20 diameter | 0–10 |
| Pl (labradorite)—m | | 3 |

**Table 1.** *Cont.*

| 1000–1700 m, Tuff/Tuffites—Linked to the Campanian Ignimbrite Zeolitized Tuff | | | |
|---|---|---|---|
| Features: phono-trachytic ashy-to-sandy tuff with pumices and scoriae | | | |
| | | Variability | Model Input Data |
| Porphyritic with matrix (%) | | 86–88 | 86 |
| **Primary Minerals:** | **Size (μm)/Type** | **wt %** | **wt %** |
| Pl (bytownite or oligoclase)—f | 400 large:700 long | 5 | 0 |
| K-Feld (adularia and sanidine)—m | 10 large:150 long | 0–51 | 0 |
| K-Feld—f | 500 large:1000 long | 5 | 5 |
| CPX (diopside)—f | 300 large: 500 long | 4–1 | 4 |
| BT—f | 200 large: 400 long | 1 | 1 |
| Glass | | 80–10 | 80 |
| ZE (phillipsite>chabazite)—m | 5 large: 5 long | 0–20 | 0 |
| Pl (bytownite or oligoclase)—m | 50 large:100 long | 5 | 10 |
| CPX (diopside)—m | 50 large:70 long | 1 | 0 |
| Anl—m | 20 diameter | 0–10 | 0 |
| 1200 to 2450 m, Lavas—Analogue: Old Lavas in Outcrops and MF1 1500, MF5 2222 Cores | | | |
| Features: porphyritic and holocrystalline groundmass | | | |
| | | Variability | |
| Porphyritic with matrix (%) | | 92–77 | |
| **Primary Minerals:** | **Size (μm)/Type** | **wt %** | |
| K-Feld—f | 400 large:700 long | 6 | |
| K-Feld—m | 10 large:150 long | 50–30 | |
| Glass | intercrystalline | 30–10 | |
| MT—m | 50 | 1 | |
| SD—f | 300 | 1–10 | |
| Pl (labradorite)—m | 10 large:150 long | 10–20 | |
| CPX (diopside)—f | 200 large:300 long | 1–7 | |
| CPX (diopside)—m | 50 large:100 long | 1–16 | |

Pl plagioclase; K-Feld alkali feldspar; CPX clinopyroxene; BT biotite; Anl analcyme; ZE zeolite; SD sodalite; SM smectite; f phenocryst; m microlite. For zeolites: phillipsite at 80% and chabazite at 20%. Mineral percentage and sizes are derived by literature data for outcropping rocks (see text) and by XRD (Figure 2 and Table S1) and new EDS-BSEM analyses on cores (see Section 4.1 and Figures 3–6). The Campanian Ignimbrite zeolitized tuff is the Campanian Ignimbrite formation modeled in this study (see Section 5.2.1) "Model Input Data" is the condition adopted in 0D-geochemical simulations.

### 5.1.1. Reservoir Stratigraphy and Mineralogy

The stratigraphic reconstruction (Tables S1 and 1; Figure 2) was based on the recovered subsurface, new core and surface (e.g., [5,8,10,11,20,21,57]) data, and field experience as well. It differs from previous ones that did not know the 14.9 ka Neapolitan Yellow Tuff eruption [20,58] and the Gauro Tuff having 14.4 ky [57]. As a whole, it made four main lithological formations, as reported below.

The first 200 m (not described by AGIP) should consist of incoherent pyroclastic deposits belonging to La Starza and to explosive eruptions at the least up to the Gauro Tuff [7,20,21]. Actually, the San Vito 1 well is nearby to the volcanic edifices (Gauro, Montagna Spaccata, Fossa Lupara, Cigliano, Astroni, Solfatara and Agnano Monte Spina) and within the dispersion area of most deposits of the recent volcanism younger than 14.4 ka (e.g., [21,57]). Furthermore, the Gauro Tuff forms the largest cone visible just to the west of the San Vito 1 well and the deposits were reached at −40 m down to >−20 m in a borehole located 2 km NW from San Vito 1 [21]. The deposits are mostly ashy-to sandy with a variable percentage of pumices and scoriae (e.g., Solfatara, Agnano-Monte Spina, Pomici Principali, Astroni, Cigliano, Montagna Spaccata, etc.), typically trachy-phonolite and rarest latites. Primary crystals are dominated by K-feldspar (hereafter, Na is generally present) and subordinately plagioclase and pyroxene; biotite, magnetite and apatite may occur, whereas olivine is rarest ([5,59,60] and references therein). Several deposits are

zeolitized by phillipsite and chabazite [61]. They are from low porphyritic (<10–20 wt %), with crystals that may reach a size of several millimeters in a glassy groundmass, to a high abundance of microcrysts and microlites (70–80 wt %) in the groundmass. In the hydrothermal area located a few kilometers east from San Vito 1, the deposits are altered; alunite, kaolinite, quartz, pyrite and native sulfur are the most widespread hydrothermal minerals [23].

The incoherent to mostly zeolitized yellow tuff downwards to ca. 1000 m is similar to other tuffs in the area; the most probable candidate is the Neapolitan Yellow Tuff [10,20,21]. The Neapolitan Yellow Tuff is largely widespread and shows a significant thickness in outcrops at the caldera rim. Accordingly, it consists of an ashy-to-sandy glassy matrix with minor juveniles that range in composition from latite to trachyte, the latite being the least abundant, the scoria upwards in the sequence, and rare lithic lava clasts [58,62]. It also characterizes for greenish elements (e.g., [10]) that should be associated to glauconite. The tuff is mostly altered in the clay (<6%) and variably zeolitized (28–62 wt %) by phillipsite, chabazite and analcyme [62–64]. The authors report sanidine phenocrysts at 14–36 wt %, amorphous at <22 wt % and minor plagioclase (dominant labradorite), pyroxene and biotite. This assemblage is similar to that observed in the SV1 804 core (see Sections 3.1.2 and 4.1) and reported by Bruni et al. (in [6]). However, at the EDS-BSEM the core appears richer in the analcyme and chlorite clays, virtually lacking of a glass phase.

The fossils-bearing tuffites and chaotic tuffs between ca. −1000 and −1660 m have been attributed to the submarine post-caldera period (≤39 ka) ([5,7,20] and references therein). In agreement with the authors, we considered that they belonged to the Campanian Ignimbrite formation. In particular, the Sr-isotope and $\delta^{18}O$ values give constraints to the correlation [5]. Moreover, the huge volume (up to 300 km$^3$) and thickness of deposits in outcrops nearby the caldera rim also support the choice. The typical ignimbrite is welded and includes trachy-phonolitic pumices and scoriae [65,66]. It is made of 80–90% matrix with sanidine phenocrysts and minor quantities (few percent) of clinopyroxene (diopside), plagioclase (bytownite to oligoclase), biotite and magnetite [66,67]. The matrix is largely devitrified (feldspatization) and zeolitized (phillipsite and chabazite), so that the tuff consists of 50–80 wt % of feldspar (adularia and sanidine) and 10–60 wt % of zeolites [66,68]. The affinity with cores at ca. 1400 m (SV1 1412, SV1 1417 and SV1 1418) consists of the scoria elements (Figure 3d), the variable crystalline lava clasts (Figure 6f), variable sized phenocrysts, several broken phenocrysts (Figures 3e and 5d) and the pyroxene occurrence (Figures 3h and S1a). K-feldspars, oligoclase to andesine plagioclases and pyroxenes are the most abundant pristine minerals. Whereas the secondary minerals are analcyme/zeolites, albite and chlorite clays. Minor calcite, quartz and pyrite (<2% in abundance) have been recovered to be dispersed mostly within voids.

The lavas intercalated in the sequence deeper than 1200 m belong to the precaldera volcanism [5,7]. They can be correlated to lavas drilled at Mofete. Punta Marmolite, Cuma and Monte Echia are a few examples of older lavas in outcrops [69]. These lavas are generally low porphyritic and consist of a highly crystallized groundmass [70]. Sanidine is the dominant phase both among phenocrysts and microcrysts. These features have been detected also in cored lavas at Mofete (see Section 4.1).

### 5.1.2. Rock Geochemistry

The composition of lithological formations in the reservoir was defined on the basis of the huge datasets existing for outcropping eruptive units [11,58,59,62,65–67,69–74]. Zeolitized tuffs are the most widespread intracaldera deposits and are therein included. Caprarelli et al. [11] provided the geochemistry of cores. Data were all normalized on a water-free basis. The initial rock composition of the reservoir units is the average rock geochemistry of the related outcropping deposits, either whole-rocks and glasses (Table 2).

**Table 2.** The chemistry of protolith rocks forming the geothermal reservoirs at San Vito 1. Rocks are grouped considering the initial reservoir stratigraphy in Section 5.1.1.

| Postcaldera Deposits Younger than the Neapolitan Yellow Tuff (Data Source: [59,73,74]) | | | | | | | |
|---|---|---|---|---|---|---|---|
| whole rock (*n*. 171) | | | | glass matrix (*n*. 462) | | | |
| Oxide | Average | Min | Max | Oxide | Average | Min | Max |
| $SiO_2$ | 58.53 | 51.39 | 63.39 | $SiO_2$ | 61.45 | 61.46 | 61.45 |
| $TiO_2$ | 0.55 | 0.38 | 0.97 | $TiO_2$ | 0.52 | 0.52 | 0.52 |
| $Al_2O_3$ | 18.49 | 14.73 | 19.98 | $Al_2O_3$ | 17.89 | 17.89 | 17.90 |
| $Fe_2O_3tot$ | 4.94 | 3.00 | 8.26 | $FeO$ | 2.45 | 2.44 | 2.44 |
| $MnO$ | 0.14 | 0.09 | 0.23 | $MnO$ | 0.14 | 0.14 | 0.14 |
| $MgO$ | 1.37 | 0.19 | 6.08 | $MgO$ | 0.96 | 0.96 | 0.95 |
| $CaO$ | 4.10 | 1.77 | 12.14 | $CaO$ | 3.44 | 3.43 | 3.42 |
| $Na_2O$ | 4.02 | 1.57 | 7.51 | $Na_2O$ | 4.32 | 4.33 | 4.33 |
| $K_2O$ | 7.64 | 3.27 | 9.32 | $K_2O$ | 8.65 | 8.66 | 8.67 |
| $P_2O_5$ | 0.23 | 0.03 | 0.63 | $P_2O_5$ | 0.18 | 0.17 | 0.17 |
| $LOI$ | 2.30 | 0.05 | 8.45 | | | | |

| Neapolitan Yellow Tuff (Data Source: [58,62]) | | | | | | | |
|---|---|---|---|---|---|---|---|
| whole rock (*n*. 52) | | | | glass matrix (*n*. 410) | | | |
| Oxide | Average | Min | Max | Oxide | Average | Min | Max |
| $SiO_2$ | 58.48 | 54.84 | 61.07 | $SiO_2$ | 59.52 | 53.71 | 63.58 |
| $TiO_2$ | 0.50 | 0.41 | 0.62 | $TiO_2$ | 0.53 | 0.40 | 0.78 |
| $Al_2O_3$ | 18.39 | 17.92 | 18.68 | $Al_2O_3$ | 18.58 | 17.42 | 19.72 |
| $Fe_2O_3tot$ | 5.41 | 3.42 | 7.76 | $FeO$ | 4.12 | 2.69 | 6.61 |
| $MnO$ | 0.13 | 0.11 | 0.15 | $MnO$ | 0.13 | 0.03 | 0.24 |
| $MgO$ | 1.06 | 0.47 | 1.96 | $MgO$ | 0.94 | 0.32 | 2.11 |
| $CaO$ | 3.58 | 2.21 | 5.73 | $CaO$ | 3.45 | 1.96 | 6.51 |
| $Na_2O$ | 3.83 | 3.18 | 5.06 | $Na_2O$ | 3.81 | 2.88 | 4.95 |
| $K_2O$ | 8.41 | 7.60 | 9.51 | $K_2O$ | 8.72 | 7.39 | 10.22 |
| $P_2O_5$ | 0.22 | 0.08 | 0.43 | $P_2O_5$ | 0.20 | 0.01 | 0.56 |
| $LOI$ | 3.24 | 1.21 | 4.49 | | | | |

| Campanian Ignimbrite (Data Source: [65,66,71,72]) | | | | | | | |
|---|---|---|---|---|---|---|---|
| whole rock (*n*. 202) | | | | glass matrix (*n*. 250) | | | |
| Oxide | Average | Min | Max | Oxide | Average | Min | Max |
| $SiO_2$ | 61.20 | 50.80 | 63.02 | $SiO_2$ | 61.94 | 58.65 | 64.89 |
| $TiO_2$ | 0.45 | 0.36 | 0.64 | $TiO_2$ | 0.40 | 0.10 | 0.93 |
| $Al_2O_3$ | 18.60 | 17.31 | 21.89 | $Al_2O_3$ | 18.90 | 17.90 | 20.61 |
| $Fe_2O_3tot$ | 3.79 | 2.73 | 5.49 | $FeO$ | 2.88 | 0.18 | 4.05 |
| $MnO$ | 0.20 | 0.06 | 0.31 | $MnO$ | 0.22 | 0.01 | 0.63 |
| $MgO$ | 0.56 | 0.20 | 2.68 | $MgO$ | 0.44 | 0.03 | 0.87 |
| $CaO$ | 2.34 | 1.26 | 14.09 | $CaO$ | 1.85 | 0.18 | 3.66 |
| $Na_2O$ | 5.23 | 1.05 | 6.86 | $Na_2O$ | 5.07 | 2.99 | 6.85 |
| $K_2O$ | 7.55 | 5.27 | 9.56 | $K_2O$ | 8.22 | 6.31 | 10.21 |
| $P_2O_5$ | 0.10 | 0.00 | 0.32 | $P_2O_5$ | 0.14 | 0.00 | 0.54 |
| $LOI$ | 2.63 | 0.41 | 9.56 | | | | |

| Intercalated Lavas (Data Source: [11,59,69,70]) | | | | | | | |
|---|---|---|---|---|---|---|---|
| lavas outcrops (*n*. 7) | | | | cored lavas (*n*. 2) | | | |
| Oxide | Average | Min | Max | Oxide | Average | core 1 | core 2 |
| $SiO_2$ | 60.33 | 59.43 | 60.87 | $SiO_2$ | 59.57 | 64.21 | 54.93 |
| $TiO_2$ | 0.57 | 0.41 | 0.72 | $TiO_2$ | 0.49 | 0.38 | 0.60 |
| $Al_2O_3$ | 25.64 | 18.39 | 31.15 | $Al_2O_3$ | 17.67 | 17.77 | 17.58 |
| $Fe_2O_3tot$ | 5.22 | 3.74 | 6.29 | $FeO$ | 4.79 | 1.99 | 7.59 |
| $MnO$ | 0.31 | 0.11 | 0.48 | $MnO$ | 0.07 | 0.06 | 0.08 |
| $MgO$ | 0.58 | 0.26 | 1.17 | $MgO$ | 2.49 | 0.77 | 4.20 |
| $CaO$ | 2.83 | 1.72 | 3.61 | $CaO$ | 5.18 | 2.04 | 8.33 |
| $Na_2O$ | 8.22 | 4.21 | 11.53 | $Na_2O$ | 4.77 | 6.70 | 2.84 |
| $K_2O$ | 9.98 | 6.72 | 13.03 | $K_2O$ | 4.79 | 6.03 | 3.56 |
| $P_2O_5$ | 0.11 | 0.06 | 0.18 | $P_2O_5$ | 0.16 | 0.05 | 0.28 |
| $LOI$ | 1.93 | 1.12 | 3.21 | | | | |

*n*. is the number of used samples, min: minimum value and max: maximum value.

### 5.1.3. Fluid Geochemistry, Temperature and Permeability

The San Vito 1 well presents an increment of the temperature from 130 to 150 °C at a depth of ca. 800 m to 420 °C at the bottom hole (Figure 2; Appendix B). This increment is mostly due to advective processes related to Na-Cl brines confined within the deeper geothermal system. In fact, Carella indicates that "wells in San Vito area are dry" and Bruni et al. stated that "the permeability is everywhere low at San Vito 1"[6], which are increasing due to fracturing in the thermometamorphic facies (Table S1). Nonetheless, fluids at a high temperature and pressure (temperature, T = 222 °C and pressure, p = 70 kg/cm at the well-head) were discharged during a purge test [6]. Such Na–Cl brines have a composition (Table 3) [28] explainable by the shallow infiltrating seawater, heated to 350 °C and chemically and isotopically modified by boiling and water–rock interactions [11] as reported in fluid geochemistry by Cioppi et al. (in [6]).

In the caldera at shallow depths several aquifers are present, discrete and diverse [6,16,75,76]. Their compositions, although variable, testify the contribution of the regional recharge (i.e., bicarbonate–alkaline–calcium–magnesium waters), of the local meteoric supply (i.e., bicarbonate–alkaline waters) and of the seawater (i.e., chloride waters). The meteoric waters coming from the northern portion of the Campi Flegrei area (TDS 0.5 $g \cdot L^{-1}$, cold T = 20 °C, of the bicarbonate type) flow underground towards the south [16] and supply (and modify) the geothermal reservoir. In the central caldera the regional recharge mix with Na-Cl brines ascending through fractures at the discharge areas [23]. Groundwater around Solfatara are high in TDS (3–33 $g \cdot L^{-1}$), rich in $Cl^-$ [16] and $Na^+$ ([32,33] and references therein), and anomalous in $H_3BO_3$, $NH_3$ and $CO_2$ content (Cioppi et al.) in [6,77,78]. At Solfatara the waters have a temperature up to 95 °C, show an acidic pH (1–5) and are enriched in $SO_4^{-2}$ (1758–3819 $mg \cdot L^{-1}$), ammonia (128–613 $mg \cdot L^{-1}$), As (up to 785 µg/L) and B (up to thousands of µg/L) [16]. They can be classified as steam-heated groundwaters determined by the local condensation of boiling vapor separated (at 31 bars and a temperature of ca. 250 °C) from the geothermal aquifer with the contribution of the local meteoric waters [12,79]. The boiling vapor from the brines injected through higher permeable areas feed the fumarole constituted aqueous fluids rich in $CO_2$ and bearing $H_2S$, $CH_4$, $N_2$, $H_2$ and CO at a temperature over 160 °C [9,16,39,80]; the temperature is 150–190 °C and 110 °C at the main Bocca Grande and Pisciarelli fumaroles, respectively (see also the monitoring data from the Osservatorio Vesuviano on the website https://www.ov.ingv.it/ov/it/bollettini) (see for examples years 2020 and 2021). The $CO_2$ reaches a flux of at least 1500 t $d^{-1}$ and a maximum value of 3000 t $d^{-1}$ [16]. Significant Hg have been also detected [81,82]. Among the available waters, those sampled at the Tennis hotel (Table 3) are considered the most representative of the shallow hydrothermal reservoir in the San Vito area [16,32,78] and were then used as a comparison to calibrate the geochemical model of Campanian Ignimbrite. AGIP data for waters discharged from the San Vito 1 and 2 wells reported incomplete geochemical analyses and too low of a pH, and thus were not considered. In Table 3, a summary of the water composition is reported, including Mofete aquifers from [28], water chemistry during the purge test at the San Vito 1 well from Bruni et al. 1983 [6] and the Tennis hotel waters from [32,36].

**Table 3.** Water chemistry at the Mofete wells, San Vito 1 well and Tennis hotel.

| | Mofete 1 | Mofete 1 | Mofete 2 | San Vito 1 | San Vito 1 | Tennis Hotel |
|---|---|---|---|---|---|---|
| Depth (m) | 550–896 | 1273–1605 | 1275–1989 | | | - |
| TDS | 30,000 | 39,500 | 18,200 | | | 4800 |
| T (°C) | 250 | 250 | 337 | | | 85 |
| pH | 7.5 | 6.5 | 6 | 3.2 | 4.38 | 7.0 |
| $SiO_2$ (mg/L) | 398 | 417 | 450 | 369 | 246 | 184 |
| Na (mg/L) | 10,025 | 12,589 | 5090 | 11,750 | 6280 | 1295 |
| K (mg/L) | 1230 | 2342 | 1180 | 8000 | 4025 | 390 |
| Ca (mg/L) | 555 | 1281 | 480 | 3290 | 1980 | 44.1 |
| Mg (mg/L) | n.d. | 5 | 1 | 1120 | 540 | 4.9 |
| Cl (mg/L) | 17,710 | 22,810 | 10,200 | 37,755 | 20,024 | 1180 |
| $SO_4$ (mg/L) | 615 | 670 | 3160 | - | - | 1136 |
| $HCO_3$ (mg/L) | 81 | 46 | 41 | 0 | 26 | 699 |
| $NH_4$ (µg/L) | n.m. | n.m. | n.m. | | | 21 |
| B (µg/L) | 125,000 | 110,000 | 140,000 | | | 32.8 |
| Li (µg/L) | 25,000 | 28,000 | 13,000 | 47,000 | 26,000 | 0.82 |
| As (µg/L) | 9000 | 11,000 | 11,000 | | | 2255 |
| Hg (µg/L) | | | | | | 4.3 |
| Ti (µg/L) | | | | | | 2.1 |
| Pb (µg/L) | | | | | | 3.3 |
| Sr (mg/L) | 34 | 41 | 14 | | | 0.5 |
| Mn (mg/L) | 7 | 17 | 25 | | | - |
| F (mg/L) | | | | 5 | 5 | 2.2 |
| Fe (mg/L) | | | | | | 0.07 |
| Al (mg/L) | | | | | | 0.55 |
| Rb (mg/L) | | | | | | 1.41 |
| $\delta^{18}O$ | 1.1 | 0.8 | n.m. | | | - |

Mofete aquifers from [28], water chemistry during the purge test at the San Vito 1 well from Bruni et al. 1983 [6] and the Tennis hotel waters from [32,36]. n.m., not measured.

## 5.2. Model Setup

In this case study, to evaluate the hydrothermal alteration of the reservoir and verify that the newly built dataset is able to capture the alteration observed in nature, we performed a 0D geochemical model constituted by a singular cubic cell (10 m side). The homogeneous porous medium is characterized by rock and mineral properties belonging to Campanian Ignimbrite deposits. This formation is the most interesting to study because of both the thickness of ca. 700 m and the alteration degree in the phyllic-chlorite/illite facies, and it is the shallowest representative reservoir formation subjected to the interaction with the fluids enriched in $CO_2$ (see Section 5.1). Moreover, the minerals defining the composition of Campanian Ignimbrite are the same of other reservoir rock formations,



although in a different amount, and we can suppose that the main fluid-rock reactions have a similar evolution over time.

### 5.2.1. Flow Conditions

The hydrological properties assigned to the rock medium are reported in Table 4. The temperature was set to 165 °C, corresponding to the average value measured at Solfatara fumaroles [13] (see Section 5.1.3) and to the condensation/evaporation level, which feeds the fluid emissions at the surface based on the data reported in the San Vito 8 well log [6]. In this log the thermometric profile shows indeed a horizontal jump from 160 to 218 °C in correspondence of some circulation losses and hint of a high permeability zone. Porosity was taken from AGIP data (Table S1), whereas density and specific heat were indirectly computed as a function of properties of all minerals forming rock with respect to their abundances as obtained by XRD analyses. Specific heat was then corrected as function of the set temperature [83–86]. Thermal conductivity was taken from the measured data on trachytes [87,88] and then corrected for temperature dependence at 165 °C [85]. Permeability (taken from AGIP data; Table S1) was included to complete the input file, but not used since the 0D model is without flow.

**Table 4.** The petrophysical rock properties of the Campanian Ignimbrite formation used for geochemical simulations.

| Density * | 2524 kg/m$^3$ |
|---|---|
| Permeability | 10 mD (i.e., ca. $10^{-14}$ m$^2$) |
| Porosity | 0.20 fraction volume |
| Thermal conductivity (dry, 165 °C) * | 1.7 W/m/K |
| Specific heat (165 °C) * | 930 J/kg/K |

*: Computed values as a weighted sum of mineral properties with respect to their abundances (Table 1) at 165 °C. Refer to Table S1 for log parameters.

The contribution of $CO_2$ at a depth was simulated by assigning a two-phase condition in the cell, with a total gas phase pressure of 8.2 bar (corresponding to a vapor pressure at 165 °C plus $CO_2$ partial pressures) and a gas saturation (Sg) of 0.2 volume fraction. The $CO_2$ partial pressure (1.19 bar) was set according to the $CO_2/H_2O$ molar ratio of 0.17 based on the measured $CO_2$ diffuse degassing rate [16,47]. Moreover, to include the significant presence of $H_2S$ in the fluids, a $H_2S$ gas phase with a partial pressure of 0.0109 bar was added in the chemical system. This value was computed considering a $H_2S/CO_2$ molar ratio of 0.009 as measured in the Bocca Grande fumaroles at the Solfatara [12,16].

To take in to account the hydrothermal weathering due to different $CO_2$ contribution at the reservoir and surface conditions, additional batch simulations were performed by varying the $CO_2$ partial pressure and temperature, as summarized in Table 5. $CO_2/H_2O$ and $H_2S/CO_2$ ratios remained unchanged in the different simulations. A temperature of 85 °C was considered in the low temperature model as measured in the Tennis hotel waters (Table 3).

**Table 5.** The list of geochemical simulations performed for Campanian Ignimbrite deposits.

| Simulation ID | Temperature, °C | Total Pressure, bar | $CO_2$ Partial Pressure, bar |
|---|---|---|---|
| 1 | 165 | 8.219 | 1.2 |
| 2 | 165 | 7.129 | 0.12 |
| 3 | 165 | 7.02 | 0.012 |
| 4 | 85 | 1.1 | 0.12 |
| 5 | 85 | 1.0 | 0.012 |

### 5.2.2. Geochemical Modeling

The initial mineral phase of Campanian Ignimbrite was defined by the extensive review described in the Sections 3.1.2 and 5.1 and new analyses in Section 4 allowing one to set which minerals are likely to dissolve (primary) and precipitate (secondary) at low and elevated $CO_2$ conditions (Table 1). The rock phase is therefore constituted by glass, K-feldspar, plagioclase and a minor amount of diopside and biotite. Since the matrix is largely zeolitized by analcyme, phillipsite and chabazite, these minerals were added, at this stage, as secondary phases. Other secondary minerals, found in cores and added to the model, are magnetite, albite, chlorite, calcite, quartz and pyrite. Moreover, to account for further secondary phases commonly associated with $CO_2$-rich fluids, we included Fe-rich illite (for which there is evidence in the drilled materials; Section 3.1.2), gibbsite and kaolinite; kaolinite is present at the discharge areas [23].

The THERMODDEM thermodynamic database [89] was employed to guarantee completeness and thermodynamic internal consistency. The biotite was modeled as a contribution of pure solid phases, i.e., annite and phlogopite. $An_{70}Ab_{30}$-plagioclase was added as a proxy for bytownite to the oligoclase series to better represent the field observations (Table 6). The clinochlore-2 mineral [90] was substituted to the original clinochlore since it better fit our model with specific site data (Table 6).

**Table 6.** Dissolution reactions and equilibrium constants (*LogK*) of solid phases added to the THERMODDEM database.

| Mineral | Dissolution Reaction | *LogK* 0 °C | *LogK* 25 °C | *LogK* 60 °C | *LogK* 100 °C | *LogK* 150 °C | *LogK* 200 °C | *LogK* 250 °C | *LogK* 300 °C |
|---|---|---|---|---|---|---|---|---|---|
| Glass | $SiAl_{0.49}Fe_{0.039}Ca_{0.032}Na_{0.16}K_{0.17}O_{2.971}$ $+1.942H^+ + 1.029H_2O$ $= H_4SiO_4 + 0.49Al^{+3} + 0.039Fe^{+2}$ $+0.032Ca^{+2} + 0.16Na^+ + 0.17K^+$ | 16.7378 | 14.8583 | 12.7166 | 10.7704 | 8.8387 | 7.2629 | 5.9046 | 4.6721 |
| $An_{70}Ab_{30}$-Plagioclase | $Ca_{0.7}Na_{0.3}Al_{1.7}Si_{2.3}O_8 + 6.8H^+ + 1.2H_2O$ $= 0.7Ca^{+2}0.3Na^+1.7Al^{+3}2.3H_4SiO_4$ | 22.648 | 18.92 | 14.421 | 10.218 | 5.995 | 2.533 | −0.474 | −3.258 |
| Clinochlore-2 | $Mg_5Al_2Si_3O_{10}(OH)_8 + 16H^+$ $= 5Mg^{+2} + 2Al^{+3} + 3H_4SiO_4 + 6H_2O$ | 76.5440 | 66.6500 | 55.1110 | 44.6070 | 34.2950 | 26.0340 | 19.0080 | 12.6210 |

LogKs are reported following in the TOUGHREACT format, i.e., at eight different temperatures (0, 25, 60, 100, 150, 200, 250 and 300 °C) and pressures (1 bar from 0–100 °C, and after vapor pressure).

The glass composition of Campanian Ignimbrite ($SiAl_{0.49}Fe_{0.039}Ca_{0.032}Na_{0.16}K_{0.17}O_{2.971}$) was defined on the basis of the datasets existing for outcropping eruptive units [65,66,69,71,72] (see Table 2, Section 5.1.2). $TiO_2$, MnO, MgO and $P_2O_5$ were neglected due to their very low content (1.20%). The obtained rock composition was recomputed as wt % and normalized with respect to one atom of Si.

Glass solubility constants (Table 6) were computed assuming that this is an oxide mixture, following the method proposed by [91] and applied by [92] for Iceland basaltic glasses. Therefore, the glass *logKs* outcome from the sum of the *logKs* of glass-constituting oxides as a function of their mole fraction is:

$$Log(K)_{glass} = \sum_i Log(K_i) + \sum_i x_i log x_i \tag{1}$$

where *xi* and *Ki* are the mole fractions and solubility products of the glass-constituting oxides, respectively.

Equilibrium constants of $An_{70}Ab_{30}$-plagioclase and clinochlore-2 were computed by the SUPCRT92 [93] code by using thermodynamic parameters provided by [90,92] respectively (Table 6).

Mineral reactions proceeded under the mechanism following the transition state theory [94–96], with the exception of kaolinite and gibbsite, which are set at equilibrium

conditions. A general equation (Equation (2)), including temperature and pH dependence of the rate constants and reaction terms, is implemented in TOUGHREACT:

$$r = S \left[ \left( \begin{array}{c} \left(k_{298.15-acid}exp\left[-\frac{E_{a-acid}}{R}\left(\frac{1}{T}-\frac{1}{298.15}\right)\right]a_{H^+}^{n_1}\right)+ \\ \left(k_{298.15-neutral}exp\left[-\frac{E_{a-neutral}}{R}\left(\frac{1}{T}-\frac{1}{298.15}\right)\right]\right)+ \\ \left(\left(k_{298.15-base}exp\left[-\frac{E_{a-base}}{R}\left(\frac{1}{T}-\frac{1}{298.15}\right)\right]a_{OH^-}^{n_3}\right)\right) \end{array} \right) \right] (1-\Omega) \quad (2)$$

where $r$ is the kinetic rate (dissolution or precipitation), $S$ is the specific reactive surface area (m$^2$ g$^{-1}$), $k_{298.15}$ is the rate constant at 298.15 K (mol m$^{-2}$ s$^{-1}$), $E_a$ is the activation energy, $R$ is the gas constant (8.314 J mol$^{-1}$K$^{-1}$), $T$ is the absolute temperature (in Kelvin), $a$ is the aqueous activity of the species, $n$ is the order of the reaction and $\Omega$ is the mineral saturation index. For precipitation only the neutral mechanism was considered.

The kinetic parameters of minerals ($k_{298.15}$, $E_a$ and $n$) were mainly taken from [97] and listed in Table 7. The data of smectite were used for Fe-illite, since they are missing. All zeolites have the same parameters taken from [92]. The glass kinetic data and dissolution mechanism are from [98]. Glass kinetic rate constants were reduced by a factor of three and its specific reactive area was decreased by a factor of two to avoid too fast of a dissolution, reduce numerical convergence problems and better fit site specific data. Similarly, pyrite kinetic rate constants were reduced by two orders of magnitude to prevent an excessive precipitation not observed in the actual mineralogical data. The use of a reduction factor was justified by the high uncertainties linked to laboratory measurements of kinetic rates and the estimation of reactive area (e.g., [99]) since only part of the mineral surface is involved in the reaction (e.g., [53]).

**Table 7.** Kinetic parameters for Equation (2) of the considered minerals.

| Mineral | Vol. % | S (cm$^2$g$^{-1}$) | K 298.15 (mol·m$^{-2}$·sec$^{-1}$) | | | Ea (kJ·mol$^{-1}$) | | | n1 | n3 |
|---|---|---|---|---|---|---|---|---|---|---|
| | | | Acid | Neutral | Base | Acid | Neutral | Base | Acid | Base |
| Diopside [a] | 0.032 | 50.98 | $4.365 \times 10^{-7}$ | $7.763 \times 10^{-12}$ | | 96.1 | 40.6 | | 0.71 | |
| Glass [b] | 0.64 | 44.44 | $4.096 \times 10^{-10}$ | | | 25.5 | | | 1.00 | $-0.33$ * |
| An$_{70}$Ab$_{30}$-plagioclase [a] | 0.08 | 370.37 | $1.349 \times 10^{-8}$ | $1.230 \times 10^{-11}$ | | | 45.2 | | 0.626 | |
| K-feldspar [a] | 0.04 | 39.06 | $8.710 \times 10^{-11}$ | $3.890 \times 10^{-13}$ | $6.310 \times 10^{-22}$ | 51.7 | 38.0 | 94.1 | 0.50 | 0.823 |
| Annite [a] | 0.00376 | 76.69 | $1.445 \times 10^{-10}$ | $2.818 \times 10^{-13}$ | | 22.0 | 22.0 | | 0.525 | |
| Phlogopite [a] | 0.00424 | 88.97 | $1.445 \times 10^{-10}$ | $2.818 \times 10^{-13}$ | | 22.0 | 22.0 | | 0.525 | |
| Albite[a] | $8 \times 10^{-7}$ | 2290.08 | $6.918 \times 10^{-11}$ | $2.754 \times 10^{-13}$ | $2.512 \times 10^{-16}$ | 65.0 | 69.8 | 71.0 | 0.457 | 0.572 |
| Analcyme [c] | $8 \times 10^{-7}$ | 5217.39 | $1.995 \times 10^{-8}$ | $1.585 \times 10^{-12}$ | $5.495 \times 10^{-15}$ | 58.0 | 58.0 | 58.0 | 0.70 | 0.30 |
| (K,Na)-phillipsite [c] | $8 \times 10^{-7}$ | 5454.55 | $1.995 \times 10^{-8}$ | $1.585 \times 10^{-12}$ | $5.495 \times 10^{-15}$ | 58.0 | 58.0 | 58.0 | 0.70 | 0.30 |
| Chabazite [c] | $8 \times 10^{-7}$ | 5741.63 | $1.995 \times 10^{-8}$ | $1.585 \times 10^{-12}$ | $5.495 \times 10^{-15}$ | 58.0 | 58.0 | 58.0 | 0.70 | 0.30 |
| Magnetite [a] | $8 \times 10^{-7}$ | 2330.10 | $2.570 \times 10^{-9}$ | $1.660 \times 10^{-11}$ | | 18.6 | 18.6 | | 0.279 | |
| Pyrite [a] | $8 \times 10^{-7}$ | 1197.60 | $3.020 \times 10^{-8}$ | $2.818 \times 10^{-5}$ | | 56.9 | 56.9 | | $-0.5$ # $0.5$ ° | $0.50$ § |
| Calcite [a] | $8 \times 10^{-7}$ | 492.00 | $0.501 \times 10^{0}$ | $1.549 \times 10^{-6}$ | | 14.4 | 23.5 | | 1.00 | |
| Quartz [a] | $8 \times 10^{-7}$ | 1132.08 | - | $3.981 \times 10^{-14}$ | - | - | 90.9 | - | - | - |
| Fe-Illite [a,d] | $8 \times 10^{-7}$ | 2181.82 | $1.047 \times 10^{-11}$ | $1.65959 \times 10^{-13}$ | $3.020 \times 10^{-17}$ | 23.6 | 35.0 | 58.9 | 0.34 | 0.40 |
| Clinochlore-2 [a] | $8 \times 10^{-7}$ | 2120.14 | $7.763 \times 10^{-12}$ | $3.01995 \times 10^{-13}$ | - | 88.0 | 88.0 | - | 0.5 | - |

Vol: initial volume fraction including the 0.2 pore fraction value (porosity). [a]: from [97]. [b]: from [98]. Rate constant reduced by 3 orders of magnitude. [c]: all zeolites have the same kinetic parameters from [92]. [d]: parameters from smectite. *: reaction order n with respect to Al$^{+3}$. #: reaction order n with respect to H$^+$. °: reaction order n with respect to Fe$^{+3}$. §: Reaction order n with respect to O$_2$(aq). S (m$^2$g$^{-1}$) was computed as the geometric area on the basis of the BSEM images and literature information (Section 5.1). n1, n3: order of the reaction for acid and base mechanisms in Equation (2).

A specific reactive area of primary minerals (Table 7) was computed as the geometric area on the basis of BSEM images and literature information (see Table 1 of Section 5.1), assuming that mineral grains are spheres, cylinders or a plate-like grain. Secondary minerals were assumed to have an initial area equivalent to spheres with a 2.5 μm radius, while a small initial volume fraction of $10^{-6}$ was assigned to initialize the minerals not included in the starting/primary assemblage.

The initial water composition of ignimbrite formation is unknown. Therefore, a NaCl (0.033 M) equivalent water with pH = 7 was used as the input data and allowed I to

equilibrate with the reservoir minerals under kinetic conditions for thousands of years. This salinity was set on the basis of stoichiometric NaCl content measured in the Tennis hotel waters (Table 3).

The initial water, mineral composition and kinetic parameters are the same in all performed simulations (Table 5).

## 6. Discussion

Deposits shallower than $-450$ m were mostly unaltered; downward the hydrothermal alteration increased with the highest in the thermometamorphic facies at the bottom well (Figure 2, Sections 3.1.2 and 4.1, Appendix B) [6]. In agreement with AGIP, the investigated cores showed secondary calcite, illite and quartz at a very low abundance of a few percent, mostly in voids, whereas the S-bearing phases were rare and occurred in the deposits at ca. 800 m of depth. The alteration essentially realized analcimization and chloritization. In particular, the analcimization appeared as a common alteration process of the argillic and phyllic facies at San Vito 1. The albitization only characterized the plagioclase richer sample (Figure 5c,d) at the limit between the argillic and the phyllitic zones, namely in the Campanian Ignimbrite formation. The processes might involve the substitution (and liberation) of Ca, K and Al by Na (supplied and lost by the circulating fluids). The albite and analcyme were not pure (Figure S2) and coexisted with the protolithic composition type suggesting that processes are achieved within certain zones of rocks and crystal grains. Corrosion fronts along the crystal rims (Figures 5a,e and 6a) and the shape of the albite patches (Figure 5d) pointed to fractures, voids and defects favoring the alteration processes where the fluids can infill and circulate. Minute inclusions or porosity were present on analcyme and absent on albite suggesting a mechanism of solid-state diffusion or dissolution-precipitation, respectively. Liberated Na, Mg and Ca by protolith products are likely used in the secondary chlorite, titanite and calcite.

The observed mineral alteration was here evaluated on the basis of preliminary 0D geochemical simulations performed at low and elevated $CO_2$ conditions and different temperatures (Table 5). The modeled fluid-rock interaction provided in fact clues on dissolution and precipitation of minerals at the Campanian Ignimbrite deposits that can be a guideline for a better understanding of the system. It also gives useful information on theoretical changes of water composition as a function of $PCO_2$ and temperature. Since the mineralogical and chemical similarity between Campanian Ignimbrite and other tephra deposits (Tables 1 and 2), the simulations have a larger application in the Campi Flegrei volcanic-field.

Batch simulations at high (165 °C) and low (85 °C) temperatures (Figures 7–10) with a variable contribution of $CO_2$ resulted in a similar evolution of mineral assemblage, indicating the likely minerals that dissolved (i.e., glass, diopside and plagioclase) and those that precipitated (i.e., K- and Na-phillipsite, analcyme, chabazite, kaolinite, gibbsite, albite, clinochlore, Fe-illite, magnetite, quartz and pyrite). Consistently with natural rocks, the predicted alteration of primary minerals evolved to zeolites and clay, which precipitate as secondary phases.

The differences among the performed simulations (Table 5) include the reaction times, extent of dissolution, amount of precipitates and so the amount of ions released in the solution. In particular, a general delay in the reaction time and a minor amount of mineral dissolved and precipitated were observed as a $PCO_2$ and temperature decrease. This behavior was expected since the kinetic rates of modeled minerals were higher in the acid environment and depend on temperature, following the Arrhenius equation. Moreover, also field stability of some solid phases was affected by temperature (e.g., gibbsite).

Although the $PCO_2$ was initialized at values reported in Table 5, fluid-rock reactions consumed the main part of $CO_2$ within the first 62 years in all the simulations (Figure 7). Therefore, hereafter only results of the first 1000 years of simulations will be showed and discussed. To a better diagram display, simulation 2 ($PCO_2$ = 0.12 bar) results will be not

shown because it had an intermeddle behavior between simulations 1 and 3, as shown in Figure 7.

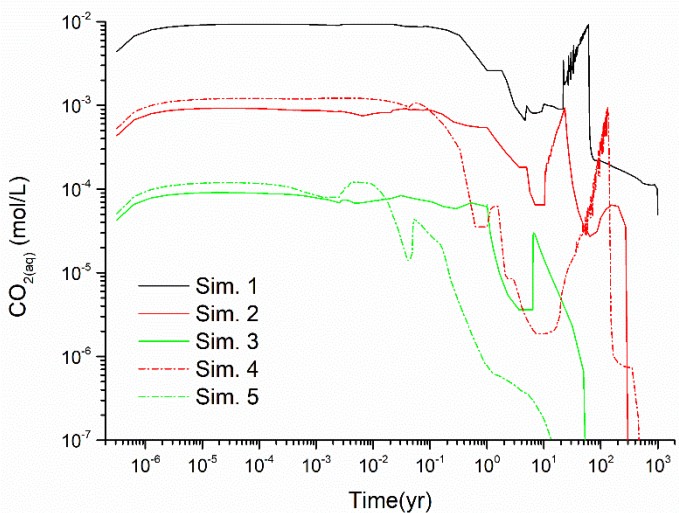

**Figure 7.** Simulations 1–5: (1) $CO_{2(aq)}$ evolution (mol/L) over time at 165 °C and $PCO_2$ = 1.2 bar (Sim. 1, black line), (2) $PCO_2$ = 0.12 bar (Sim. 2, red line), (3) $PCO_2$ = 0.012 bar (Sim. 3, green line), (4) at 85 °C and $PCO_2$ = 0.12 bar (Sim. 4, red dot line) and (5) $PCO_2$ = 0.012 bar (Sim. 5, green dot line).

In all the simulations the Campanian Ignimbrite reactions were mainly controlled by glass, $An_{70}Ab_{30}$-plagioclase and diopside dissolution. The glass dissolution (not shown) progressively increased over time for all simulations, while diopside alteration was higher and faster for $PCO_2$ = 1.2 simulation (sim. 1 in Table 5) showing a braking after 62 years as the available $CO_2$ was consumed (Figure 8a). $An_{70}Ab_{30}$-plagioclase displays a different trend depending on the $CO_2$ content. For $PCO_2$ higher than 0.4 bars (simulations 1 and 2) it dissolved, whereas somewhere between 0.4 and 0.3 bar of $PCO_2$ it weakly precipitated (e.g., simulation 3 in Figure 8a). This interval allowed one to define a transition zone between the rock portion affected by the uprising of hydrothermal $CO_2$-rich fluids and the rest of the reservoir. A similar behavior was replayed in simulations 4 and 5 (Figure 9a), with the $An_{70}Ab_{30}$-plagioclase dissolving for $PCO_2$ = 0.12 bar and remaining quite stable for $PCO_2$ = 0.012 bar.

The primary micas (annite and phlogopite; not shown) stayed essentially unchanged over time in all 1–5 simulations, whereas K-feldspar minimally precipitated.

The incongruent dissolution of primary minerals releases cations in solutions, which can precipitate as secondary minerals, in particular as zeolites and clays. In the high temperature simulations, the precipitation reflects the cation amount in the glass composition, which is the main component in the selected protolithic rock composition (Table 1). Therefore, the most abundant formed zeolites were K-phillipsite, analcyme and chabazite (Figure 8b). Na-phillipsite (Figure 8b) and albite (Figure 8d) precipitated early but completely dissolved after about 1100 years, likely due to competition with analcyme for the $Na^+$ released by glass and $An_{70}Ab_{30}$-plagioclase dissolution. In the low temperature simulations, more Na-phillipsite and chabazite precipitated over analcyme (Figure 9b), but again Na-phillipsite and albite completely dissolved after about 16,000 years.

Calcite (Figures 8c and 9d) formed in the first years of simulations and remained stable as $CO_2$ was consumed and the pH exceeded the 8.2 value, just for the $PCO_2$ higher than 0.12 bar (simulations 1, 2 and 4). In simulations 3 and 5 the calcite did not precipitate, likely due to the minor amount of $HCO_3^-$ in the solution and the competition with chabazite and $An_{70}Ab_{30}$-plagioclase for the available $Ca^{+2}$ in the solution.

The performed simulations correctly modeled the formation of clay minerals as Fe-illite, clinochlore and kaolinite. Clinochlore rapidly increased until $CO_2$ was consumed and after rose slowly (Figures 8c and 9c), reflecting the diopside alteration. Fe-Illite

was the main clay precipitated and its abundance sensibly increased as PCO$_2$ decreased (Figures 8c and 9c). On the contrary, kaolinite quickly precipitated with high CO$_{2(aq)}$ content, but it completely dissolved after about 1000 years and 450 years in high and low temperature simulations, respectively (Figures 8d and 9d). Its formation and that of gibbsite were linked to Al$^{+3}$ availability and imposed thermodynamic equilibrium. The different temperature in simulations affected the gibbsite behavior. At 165 °C this mineral gradually formed, but totally dissolved from 26,000 and 734 years, depending on PCO$_2$. At 85 °C, gibbsite continuously precipitated as this mineral was more stable at lower temperatures than kaolinite.

Fe$^{+2}$, released by glass dissolution, mainly precipitated as Fe-illite and magnetite (Figures 8c and 9c). Pyrite (not shown), which was gotten slower by reducing its kinetic constants by a factor of two, minimally formed only for a PCO$_2$ higher than 0.4 bar (simulations 1 and 2) and after it remained stable, whereas it did not precipitate in the other simulations. Finally, quartz (not shown) definitely precipitated and dissolved in the first 50 years of simulation.

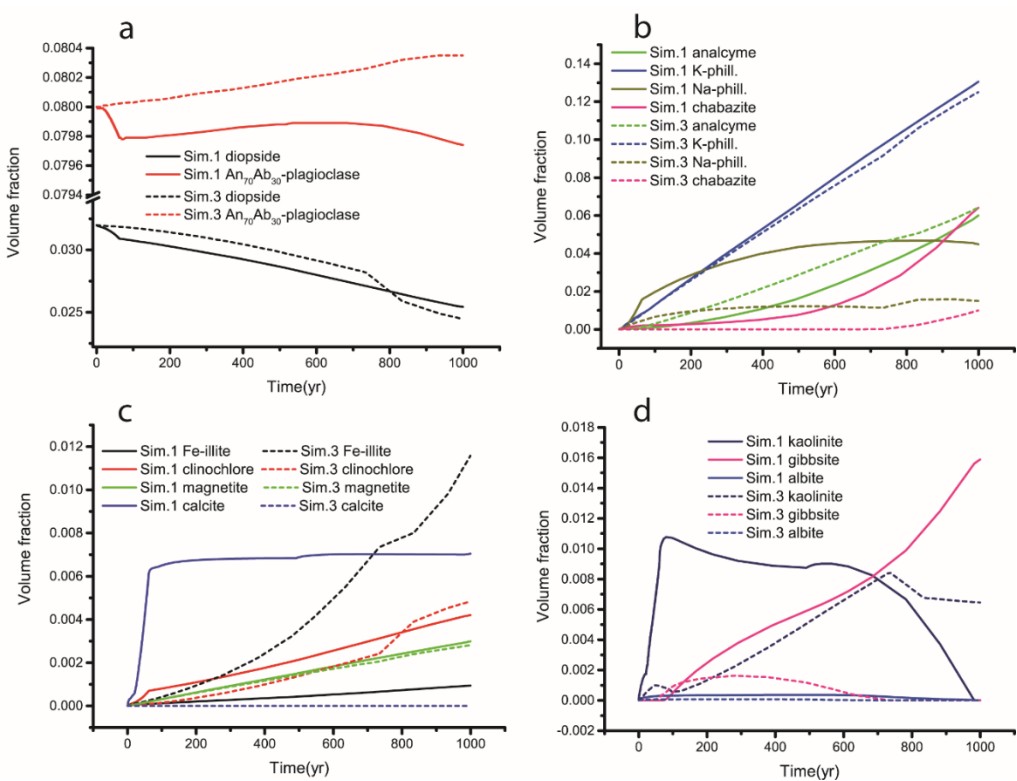

**Figure 8.** Simulations 1 and 3: mineral evolution (fraction volume) over time at 165 °C and PCO$_2$ = 1.2 bar (Sim. 1, solid lines) and PCO$_2$ = 0.012 bar (Sim. 3, dash lines). Initial volume fraction as reported in Table 7. (**a**) Primary diopside and An$_{70}$Ab$_{30}$-plagioclase; (**b**) secondary zeolites; (**c**) secondary Fe-illite, clinochlore, magnetite and calcite; (**d**) secondary kaolinite, gibbsite and albite.

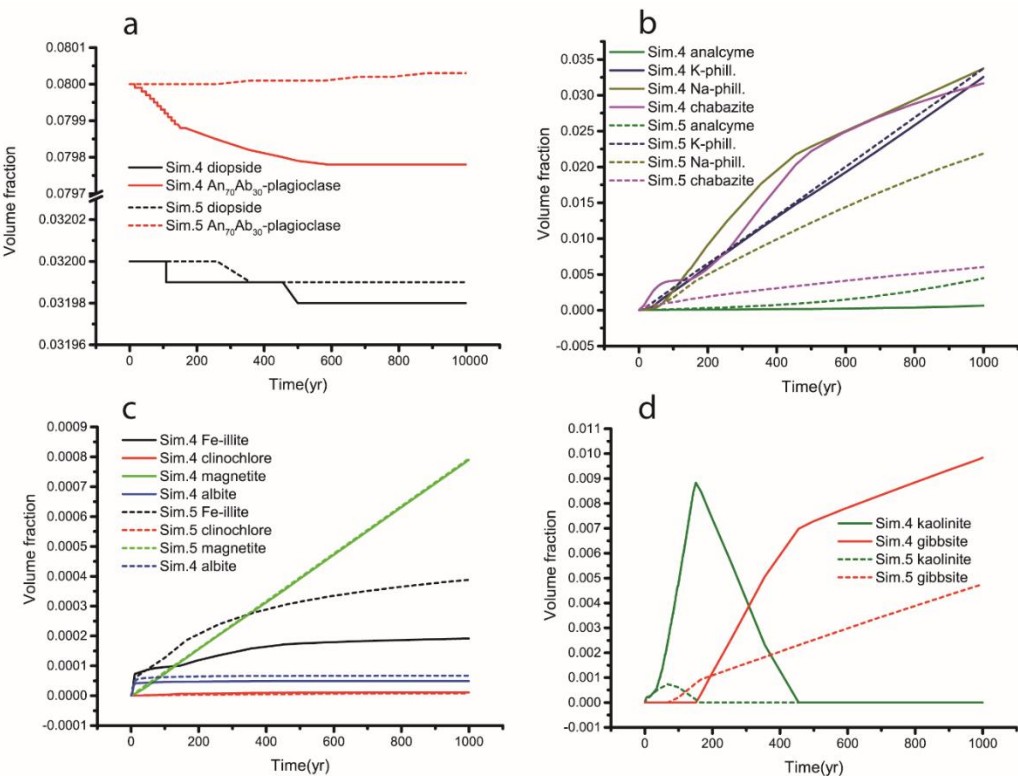

**Figure 9.** Simulations 4 and 5: mineral evolution (fraction volume) over time at 85 °C and $PCO_2 = 0.12$ bar (Sim. 4, solid lines) and $PCO_2 = 0.012$ bar (Sim. 5, dash lines). Initial volume fraction as reported in Table 7. (**a**) Primary diopside and $An_{70}Ab_{30}$-plagioclase; (**b**) secondary zeolites; (**c**) secondary Fe-illite, clinochlore, magnetite and albite; (**d**) secondary kaolinite and gibbsite.

The natural mineralogy of the San Vito area, if compared to the high temperature simulation outcomes, seemed to result in the interaction between primary rock and fluids with a $PCO_2$ ranging from <1.2 and >0.3 bars. Indeed, simulation 1 ($PCO_2 = 1.2$) produced a precipitation of a relatively high amount of gibbsite in the first 1,000 years that were not supported by investigations on the core material. On the contrary, $PCO_2 < 0.4$–0.3 bars (simulations 2 and 3) determined $An_{70}Ab_{30}$-plagioclase precipitation, which was partially observed as the albitization process; even if we could not exclude that this phenomenon should disappear by considering different plagioclase compositions. Low $PCO_2$ simulations (i.e., simulations 2 and 3) better agreed with the natural observations also for kaolinite, giving a limited precipitation of this phase that is usual at the surface but not in the depth of the geothermal system. The overprecipitation of kaolinite and gibbsite with respect to the observed data could be due to the assumption of the equilibrium reaction of these minerals, which accelerated their formations in the first 1000 years, although they completely dissolved in the longer period. The equilibrium condition was due to the need of reducing the excessively high $Al^{+3}$ concentrations in the solution and favoring model convergence. This $Al^{+3}$ overabundance may in turn derive from an overestimation of initial glass abundance (Table 1) and its dissolution kinetic rate (Table 7). The initial glass can be reduced considering phillipsite and chabazite zeolites among protolith phases as observed in the outcropping deposits (Section 5.1.1). Moreover, it is important to note that in batch modeling, dissolution and/or precipitation processes can be amplified by the lack of fresh fluid recharge since the model disregards the flow. The adding of secondary alunite ($KAl_3(SO_4)_2(OH)_6$) instead of gibbsite in preliminary simulations did not solve the problem since this mineral did not form both under equilibrium and kinetic condition, also due to the low $H_2S$ content.

In agreement with the actual observations, simulations restituted a low abundance of secondary calcite, pyrite and sulfates and the widespread clinochlore, as well. However,

the model overestimated Fe-illite over clinochlore formation, which is observed in the only drilled material. This disagreement is due to the use of clinochlore (i.e., Mg-chlorite) instead of a Fe-Mg-phase in the model, which leads to a precipitation of a Fe-clay to balance $Fe^{+2}$ in the solution.

Some other overprecipitations with respect to the observed data were mainly due to the employment of pure (end-member) solid phases in the database. This is the case of K-phillipsite, which is observed in the outcropping Campanian Ignimbrite deposits, while for the core material, analcyme is the dominant zeolite. This could be due to the available thermodynamic data for these minerals [89,100], which lead to its easy formation in systems rich in $CO_2$ and volcanic rocks or due to other processes observed in zeolites (e.g., the non-ideality and cation exchange mechanisms) not reproduced in our model. Similarly, the simulated precipitation of K-feldspar, which is very limited in reality, could be due to a lack of the mixed K-Na phase (i.e., anorthoclase $(Na-K)AlSiO_3$) in the model other than being due to too fast of a dissolution of glass. Indeed, the observed K-feldspar generally contained Na rather than K and resulted in often being weathered by analcyme.

Finally, titanite ($CaTiSiO_5$) was recognized as a secondary mineral at the San Vito well. The choice of not including this mineral in the model was because of its very low content (0.40 wt %) that was due to the chabazite overabundance in the simulation results.

Considering the limited discrepancies between the observed and modeled data, we compared the simulated (simulations 1–3) and Tennis hotel water chemistry (Table 3) to assess the soundness of the thermodynamic model. To test our hypothesis that the San Vito natural mineralogy seems to be result of an interaction between the primary rock and fluids with a $0.3 < PCO_2 < 1.2$ bars, we charted in Figure 10 also a result from a $PCO_2 = 0.4$ bar and 165 °C simulation for the comparison (Figure 10).

The correlation of our model with this water was used just as a reference to evaluate the reaction times and the equilibrium degree of fluids with the rock without demanding an exact correspondence. Water chemistry resulting from the modeled fluid-rock interaction can be used for comparison with the composition of the other aquifers elsewhere in the area. Since the used mineral association and chemistry of the Campanian Ignimbrite deposits (Tables 1 and 2) were similar to other tephra, the simulations had a larger application in the Campi Flegrei volcanic-field. Therefore, the changes in the rock and water composition was evaluated also at lower temperatures (85 °C; simulations 4 and 5; Figure 11) characterizing the acid-sulfate setting at the surface discharge area.

In the high temperature model, the concentration of ions and neutral species in solution (Figure 10) was generally directly related to $PCO_2$, i.e., the higher the $PCO_2$, the higher the ion concentrations in the solution. This is true for all the considered species except for $Al^{+3}$ and partially for $Na^+$ and $K^+$. The best matches between the real and simulated concentrations were achieved by $PCO_2 = 1.2$ bar simulations 1 and $PCO_2 = 0.4$ bar and partially by $PCO_2 = 0.12$ bar simulation 2. By contrast, $PCO_2 = 0.012$ bar simulation 3 shows too low or too high values, depending on the considered species. In particular, $Ca^{+2}$, $Mg^{+2}$ and $HCO_3^-$ values (Figure 10a–c) fit well the reference ones within the first 3–10 years for simulations 1 and $PCO_2 = 0.4$ bar, suggesting that $CO_2$ and diopside dissolution were correctly modeled. After this time the concentrations of $Ca^{+2}$ and $Mg^{+2}$ quickly declined due to calcite, chabazite and clinochlore precipitation. All simulations agreed well with the Tennis hotel sample for $Na^+$, $K^+$ and $H_4SiO_4$ (Figure 10d–f) between 5 and 200 years, although they slightly overestimated (up to four times higher) the $K^+$ and $H_4SiO_4$ values, due to minor Fe-illite precipitation in simulation 1 and $PCO_2 = 0.4$ bar and a minor quartz formation in $PCO_2 = 0.012$ bar simulation 3.

Computed $Fe^{+2}$ (Figure 10g) tended to be lower than that of the Tennis hotel sample for all the simulations, although an agreement could be found for different time spans. Simulations 1 and $PCO_2 = 0.4$ bar better guessed the expected values in the first day and at 62 years, $PCO_2 = 0.12$ bar simulation 2 always underestimated, whereas $PCO_2 = 0.012$ bar simulation 3 better fit between 80 and 2300 years. The differences between calculated and actual values could be attributable to a $Fe^{+2}$ overestimation in the analytical data due

to the iron colloids presence and/or major precipitation of magnetite and Fe-illite in the model, due to, as stated above, the employment of Fe-illite instead of Fe-Mg-chlorite. $Al^{+3}$ concentrations (Figure 10h) increased as $PCO_2$ decreased and followed the pH behavior (Figure 10i), which controlled $Al^{+3}$ solubility. Simulation 1 better fit the Tennis hotel concentration until the available $CO_{2(aq)}$ was consumed. The pH was not an easy parameter to compare, since the sampling at the surface was subjected to degassing. Computed pH in all simulations was generally higher than the reference value. This could be due to major buffer reactions in the model with respect to the real processes. However, the high pH coupled with the computed $SO_4^{-2}$ (Figure 10j), which was more than three orders of magnitude lower than the concentration measured at the Tennis hotel. This mismatch could be due to the presence of a shallow source of sulfur, which enriches the Tennis hotel waters in sulfates rather than the contribution of a deep origin fluid. Indeed, to obtain values comparable to the actual data we should increase the $PH_2S$ in the model of two orders of magnitude, which is not confirmed neither in the precipitated sulfur-species or in the present values measured at fumaroles. The $H_2S/CO_2$ contents higher than those used in this study (i.e., 0.019 vs. 0.009) were only measured during the main unrest episode in 1982-1982 [39]. A shallow source of sulfurs that dissolves and oxidizes in the water lowering the pH and increasing $SO_4^{-2}$ could be corroborated by the sulfides and sulfates presence in the downhole (Table S1). Shallow sulfur could also arise from the acidic sulfate setting in which the Tennis hotel water occurs; in that case the acidic setting would be considered an old or exhausting (not more or a limited supply from sulfur of deep sources) environment. The most likely possibility is that a shallow sulfur deposit, formed during many years of activity, is now dissolving due to the migration of the main activity. It is indeed a common phenomenon in many volcanic systems to have a large sulfur deposit related to the hydrothermal system and evidence of such deposits were found at La Solfatara by e.g., [37].

It is interesting to note that for all considered aqueous species, the best fitting between the computed and actual concentrations was (contemporaneously) reached within the first 10 years of simulation. This suggests that the reaction and fluid uprising times are quite fast. The high velocity of fluids circulation at Campi Flegrei is confirmed by the investigations on fluid degassing at fumaroles and field measurements of diffuse degassing in the Solfatara area. For fumaroles and from Bocca Grande, Aiuppa et al. and Tamburello et al. [17,101] reported flows between 150 and 500 t/day. These values devised for the sampling area and vapor density provide a fluid velocity ranging from 0.024 to 4.6 m/s. Similarly, $CO_2$ exhalation fluxes from soils measured by Aiuppa et al. and Chiodini et al. [17,47] give a velocity ranging from 0.0042 to 0.02 m/s, thus obtaining from a few hours to some days to migrate from 1700 m to the surface.

Low temperature simulations (simulations 4 and 5) were performed at 85 °C, the same for the Tennis hotel waters, and a variable contribution of $CO_2$, i.e., $PCO_2$ = 0.12 and 0.012, with a vapor pressure of 1 bar (Table 5). As before, in this model, the ions concentration in the solution (Figure 11) were related to the $PCO_2$ value and generally increased with it. The slower mineral kinetic due to a lower temperature suggests that equilibrium between aqueous and solid species was not reached yet even after 1000 years. Simulations 4 and 5 underestimated the concentrations of $Ca^{+2}$, $Mg^{+2}$, $HCO_3^-$ and $SO_4^{-2}$ with respect to the Tennis hotel values, suggesting that higher $PCO_2$ and $PH_2S$ were needed to match the expected values. On the contrary, the computed $Na^+$, $Al^{+3}$, $K^+$ and $H_4SiO_4$ and partially $Fe^{+2}$ concentrations agreed with the reference values only in a time span ranging from 20 to 330 years suggesting the silicate reactions controlling these species abundance are not strictly dependent on $PCO_2$ and temperature. The low temperature model correctly computed the dissolved $CO_2$, which was higher than the high temperature model (Figure 7). However, the buffer reactions quickly consumed it leading to a pH higher than 1–3 simulations.

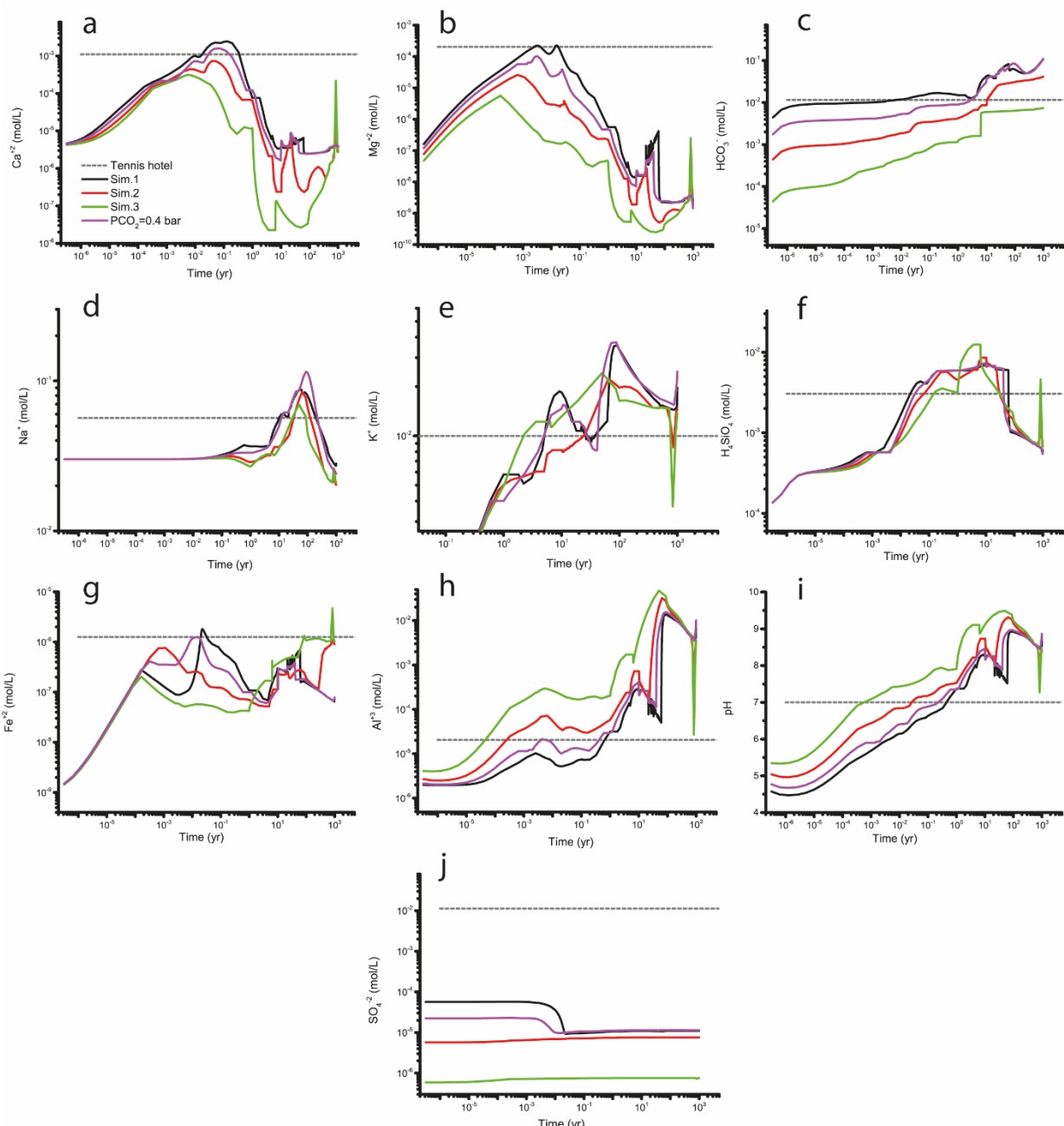

**Figure 10.** Simulations 1, 3 and $PCO_2$ = 0.4 bar: an aqueous species evolution over time at 165 °C, $PCO_2$ = 1.2 bar (black lines), $PCO_2$ = 0.12 bar (red lines), $PCO_2$ = 0.012 bar (green lines) and $PCO_2$ = 0.4 bar (purple lines). Simulation results were compared to the geochemical analysis of the Tennis hotel (gray dash lines; Table 3). (**a**) $Ca^{+2}$; (**b**) $Mg^{+2}$; (**c**) $HCO_3^-$; (**d**) $Na^+$; (**e**) $K^+$; (**f**) $H_4SiO_4$; (**g**) $Fe^{+2}$; (**h**) $Al^{+3}$; (**i**) pH and (**j**) $SO_4^{-2}$. The initial water composition of simulations was set as a NaCl (0.033 M) equivalent water, with neutral pH and initial concentrations of $1 \times 10^{-9}$ mol/L for all the other ions and neutral species.

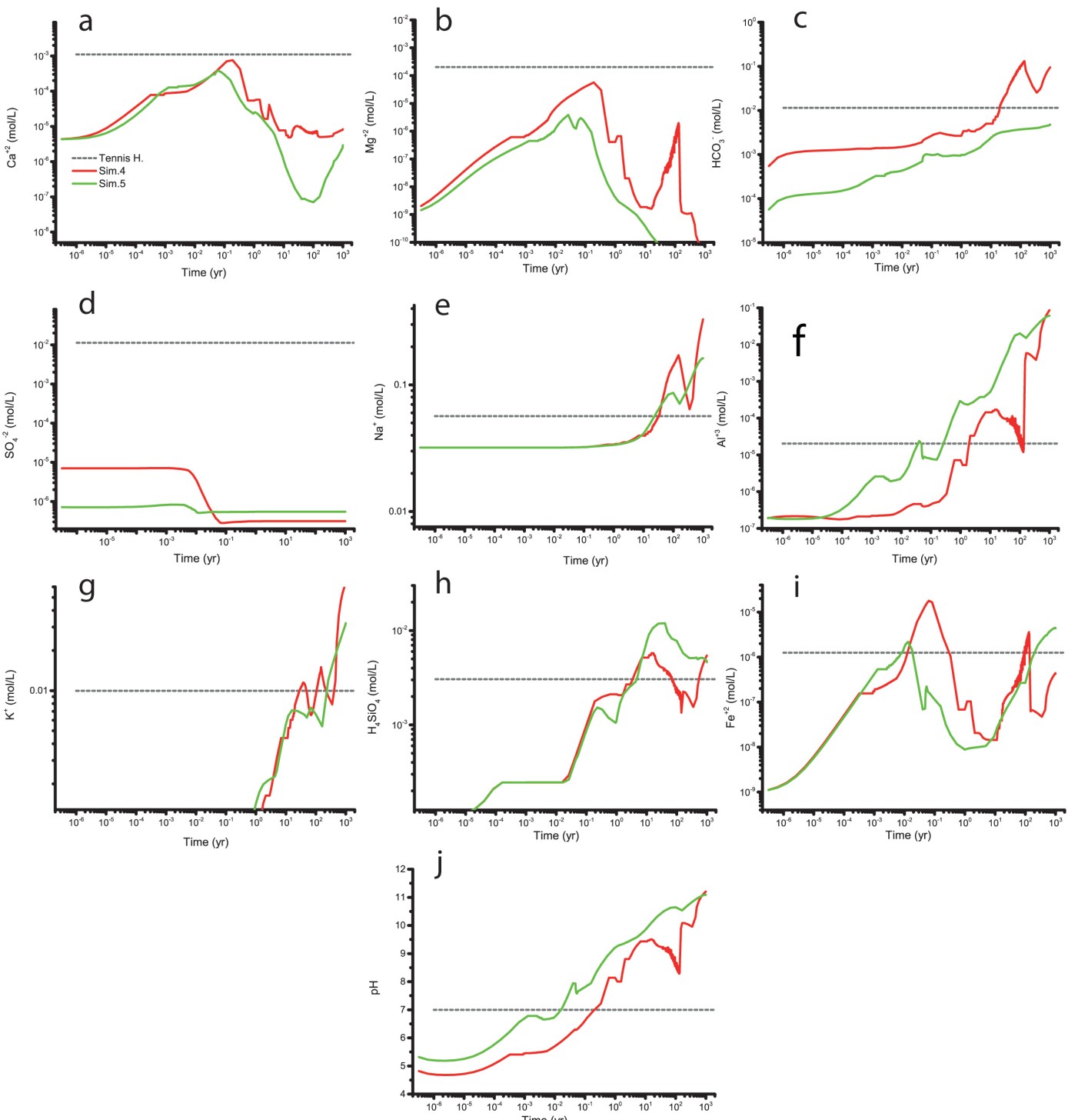

**Figure 11.** Simulations 4 and 5: an aqueous species evolution over time at 85 °C and PCO$_2$ =0.12 bar (red lines) and PCO$_2$ = 0.012 bar (green lines). Simulation results were compared to the geochemical analysis of the Tennis hotel (Table 3). (**a**) Ca$^{+2}$; (**b**) Mg$^{+2}$; (**c**) HCO$_3$$^-$; (**d**) SO$_4$$^{-2}$; (**e**) Na$^+$ (**f**) Al$^{+3}$; (**g**) K$^+$;(**h**) H$_4$SiO$_4$; (**i**) Fe$^{+2}$; (**j**) pH. Simulated initial water composition was set as a NaCl (0.033 M) equivalent water, with neutral pH and initial concentrations of 1 × 10$^{-9}$ mol/L for all the other ions and aqueous species.

## 7. Conclusions

This paper provides an integrated mineralogical and thermodynamic database useful to constrain future reactive transport simulations at the Campi Flegrei geothermal system.

Available data and new petrographic analyses were rearranged in a conceptual model useful to define representative geochemical and petrophysical parameters of reservoir rock formations suitable for 3D numerical simulations. The built geochemical model and thermodynamic database were then calibrated and validated with actual data by the 0D geochemical simulations, performed at low and elevated $CO_2$ conditions and different temperatures. The performed model correctly reproduced the alteration of primary minerals and precipitation of zeolites and clay as secondary phases, although some discrepancy due to the assumptions and available data constraints were present.

Two main outcomes were achieved. The first suggests that the observed mineral alteration is due to the fluid-rock interaction with $PCO_2$ ranging from 1.2 and 0.4–0.3 bars. Simulations within this $PCO_2$ interval better reproduced the dissolution of $An_{70}Ab_{30}$-plagioclase and agreed well with the reference aqueous concentrations sampled at the Tennis hotel. The $An_{70}Ab_{30}$-plagioclase shows indeed a double behavior, dissolving for high $PCO_2$ and precipitating for low values. This behavior could allow one to define a transition zone between the rock portion affected by the uprising of hydrothermal $CO_2$-rich fluids and the rest of the reservoir.

The second outcome concerns the fluid migration velocity. Indeed, the simulated aqueous concentrations best matched the reference Tennis hotel values within 10 years of simulation, hinting that the fluid-rock reactions and the fluid circulation toward the surface are quite fast. This process is confirmed also by the high gas flows sampled at fumaroles and from soil exhalation.

The results of this study are relevant, although preliminary, to understand the hydrothermal circulation of heat and fluids that moves from recharge zones toward sulfate discharge areas, sustained by the surplus of heat released by magma bodies in the Campi Flegrei area. We expect that future modelings can be used to simulate geophysical and geochemical signals and aid in depicting plausible evolutionary scenarios of the volcano dynamics.

**Supplementary Materials:** The following are available online at https://www.mdpi.com/article/10.3390/min11080810/s1, Table S1: Lithology, petrophysical parameters, temperature, XRD-derived primary and secondary mineralogy, alteration facies along the San Vito 1 well. Figure S1: The cored reservoir rocks cored at San Vito 1 (**a,b**) and Mofete 1 (**c**) wells under the optical microscope. (**a**) The chaotic tuffs (representing the modeled Campanian Ignimbrite formation) cored at depths of 1412 m from the phyllic facies. The image displays, under the cross-polarized light, the same area in Figure 3g. It shows a lava clast that occurs within the 1412 m chaotic tuff and that exhibits a pyroxene phenocrysts set in a matrix strongly microcrystallized by mostly plagioclase. (**b**) The chaotic tuff (representing the modeled Campanian Ignimbrite formation) cored at 1713 m of depth from the propilitic - calcium-aluminium silicate facies in the San Vito 1 borehole. The image shows the low vacuolar and locally crystalline matrix under the plane-polarized; it presents the appearance of the widespread calcite set in the inter-grains. (**c**) The lavas cored at depths of 1500 m from the phyllic facies in the Mofete 1 borehole. The plane-polarized image shows the highly crystalline matrix with several percent of sulfides and voids-filling calcite represented in Figure 4b; the plagioclase microcrysts are altered. Arrows indicate feldspar-rich portions. Abbreviations: px, pyroxene; pl, plagioclase; feld, feldspar; cc, calcite. Refer to Figure 2a for downhole location and aspect of cores. Additional details are in the text. Figure S2: EDS spectra and semi-quantitative composition of mineral phases in rock cores from the San Vito 1 well in the Campi Flegrei geothermal system.

**Author Contributions:** Conceptualization, M.P., B.C., G.C. and G.M.; methodology, M.P., B.C., G.M. and G.C.; software, B.C. and G.C.; formal analysis, M.P. and B.C.; investigation, M.P. and B.C.; resources, M.P.; data curation, M.P.; writing—original draft preparation, M.P. and B.C.; writing—review and editing, M.P., B.C. and G.M.; project administration, G.C. and M.P.; funding acquisition, M.P. and G.C. All authors have read and agreed to the published version of the manuscript.

**Funding:** This research was funded by the Istituto Nazionale di Geofisica e Vulcanologia, Italy, grant Progetto INGV Pianeta Dinamico (code CUP D53J19000170001) thanks to MIUR ("Fondo Finalizzato al rilancio degli investimenti delle amministrazioni centrali dello Stato e allo sviluppo del Paese, legge 145/2018)-Task V2-2021.

**Data Availability Statement:** Not applicable.

**Acknowledgments:** M.P. is particularly grateful to Claudia Troise and De Natale of the Istituto Nazionale di Geofisica e Vulcanologia (INGV), sezione di Napoli, Osservatorio Vesuviano, Italy, for allowing her first experience of geothermal exploration activity. M.P. expresses gratitude to Claudia Troise for the portion of cores she donated. M.P. also gives thanks Vilardo Giuseppe of the Laboratory of Geomatics and Cartography at the INGV for the DTM in Figure 1 and Angela Mormone of the INGV for kindly discussing the optical microscope. The INGV electron microscope laboratory at the Osservatorio Vesuviano have been financially supported by the EPOS Research Infrastructure through the contribution of the Italian Ministry of University and Research (MUR). The authors acknowledge two anonymous reviewers for constructive comments and editorial staff for handling the manuscript.

**Conflicts of Interest:** The authors declare no conflict of interest.

## Appendix A

(1)   This appendix lists the references that are included in the AGIP's report and that are not available in the most common search channels of scientific literature.

(2)   Antrodicchia, E.; Cioni, R.; Chiodinim G.; Gagliardi, R.; Marini, L. Geochemical temperature of the thermal waters of Phlegraean Fields (Naples, Italy). Geothermal Resources Council Transactions, 1985, Vol. 9-Part. I.

(3)   Balducci, S.; Chelini, W.; Ottonello, G. Hydrothermal equilibria in the active Mofete geothermal system (Phlegrean Fields). Fifth Symposium Water-Rock Interaction, August 1986, Reykjavik -1celand, pp. 8–17.

(4)   Bruni, P.; Sbrana, A.; Silvano, A. Risultati geologici preliminari dell'esplorazione geotermica nell'area dei Campi Flegrei. Rend. Soc. Geol. It. 1981, Vol.4, pp. 231–236.

(5)   Bruni, P.; Chelini, P.; Chelini, W.; Sbrana, A.; Verdiani, G. Deep exploration of the S. Vito Area-Pozzuoli-Na, Well S. Vito lV. Third International Seminar, European Geothermal Update Munich 29 November–l December 1983, pp. 390–406.

(6)   Carella, R.; Guglielminetti, M. Multiple reservoirs in the Mofete Fields, Naples Italy. Ninth Workshop on Geothermal Reservoir Engineering. Stanford University, California, 13–15 December 1983.

(7)   Carella, R.; Palmerini, C.G.; Stefani, G.C.; Verdiani G. Geothermal activity in Italy: Present status and prospects. Seminar on Utilization of Geothermal energy for electric power production and space heating, Florence 14–17 May 1984.

(8)   Carella, R. Status of geothermal activities by AGIP in Italy". Geothermal Resources Council Transaction, 1985, Vol. 9-Part.1.

(9)   Chelini, W. Alcuni aspetti geologico petrografici sul sistema geotermico Flegreo. Rendiconti della Società Italiana di Mineralogia e Petrologia, 1984, Vol. 39, pp. 387–391.

(10)  Cioppi, D.; Ghelardono, R.; Panci, G.; Sommaruga, C.; Verdiani, G. Demonstration project: Evaluation of the Mofete high enthalpy reservoir (Phlegraean Fields). Commission of the European Communities, Second International Seminar-Palais des Congres, Strasbourg, 4–6 March 1980.

(11)  La Torre, P.; Nannini, R. Geothermal well location in southern Italy: The contribution of geophysical methodstt. Bollettino di Geofisica Teorica ed Applicata, 1980, Vol.XX11 n.87, pp. 201–209.

## Appendix B

This Appendix reports the hydrothermal zones and temperatures for the San Vito 1 well. Deposits shallower than $-450$ m are unaltered.

Argillitic zone (depth range: 450–800 m): prevalent analcime, montmorillonite and calcite; temperature: 130–150 °C.

Phyllitic zone (depth range: 800–1600 m): prevalent illite and chlorites, calcite, analcime and sporadic anhydrite and rare pyrite. At ca. 1.300 m it is reported "a certain silicification (opal-chalcedony) and the albitization of the plagioclases"; temperature: 150–285 °C.

Phyllitic-propylitic zone (depth range: 1600–2100 m): illite, chlorites, calcite, quartz, albite, epidote, occasional zeolites and anhydrite; temperature: 300 °C.

Propylitic-potassic zone (depth range: 2100–2600 m): illite, quartz, calcite, albite, adularia, epidote, idiomorphic scapolite and occasional tourmaline and stilpnomelane. Silification is intense; temperature: >300 °C

Biotitic-actinolitic zone (depth range: 2600–2800 m): illite, quartz, adularia, epidote, biotite, actinolite, scapolite and occasional pyrrhotine; temperature: >300 °C.

Clinopyroxenic zone (depth range: 2800–3046 m): illite, quartz, plagioclase, adularia, biotite, amphibole (orneblende), clinopyroxene (diopside), scapolite and a sporadic presence of pyrrhotine, tourmaline, garnet, titanite and apatite; temperature: 410 °C.

Garnet was detected at different levels (1715, 1900, 2000 and 2500 m) [6,52].

**Appendix C**

This appendix provides the main lithological feature and mineral occurrence that we verified by petrographic investigations of cores in Figure 2a.

804 m-core: zeolitized yellowish to reddish tuff made of a chloritized trachytic vacuolar glassy matrix with millimeter-sized pumice, scoria and lithic clasts having a lati-trachytic nature. Minerals in the 804 m-core: feldspar, plagioclase, pyroxene, biotite, magnetite, clays, chlorite, hematite, zeolite/analcime, sulfates, pyrite (rare), calcite (rare) and quartz (likely), mostly in voids.

1412 m-core: chaotic tuff containing millimeter-sized pumice, scoria and lithic clasts. Minerals in the 1412 m-core: plagioclase, feldspar, pyroxene, biotite, magnetite, hematite, clays (smectite/illite), chlorite, pyrite (rare), pyrrhotite (perhaps), muscovite, calcite and zeolite/analcime.

1417 m-core: chaotic tuff with millimeter-sized pumice, scoria and lithic clasts. Minerals in the 1417 m-core: feldspar, pyroxene, biotite, magnetite, chlorite, illite, sericite (isolated or in network), pyrite, pyrrhotite (perhaps in voids), hematite, garnet, zircon, calcite (rarely isolated and more often in clots) and zeolite/analcime.

1418 m-core: vacuolar homogeneous rocks, likely a protolithic ashy-to-sandy tuff. Minerals in the 1418 m-core: feldspar, calcite, quartz (rare in voids), biotite, chlorite, clays, sericite, pyrite and zeolite/analcime.

1713 m-core: tuff, ashy-to-sandy with low-to-highly crystalline scoria; it includes vacuolar trachytic lava clasts with feldspar acicular microcrysts. Minerals in the 1713 m-core: pyroxene, feldspar, magnetite, calcite, dolomite (rare), quartz (rare), adularia, chlorite, albite, garnet, biotite, pyrite, pyrrhotine (likely), rhodochrosite, zeolite/analcime, epidote and clays (minor illite, sericite).

2130 m-core: chaotic tuff with rounded trachytic lithic clasts. Minerals in the 2130 m-core: feldspar, quartz, clays (illite, sericite), chlorite, pyrite, pyrrhotite, adularia, pyroxene, epidote, titanite, magnetite, spinel, albite, calcite and garnet.

2514-2515-2516 m-cores gray tuff with a sandy massive matrix having a trachy-latitic nature.

2683 m-core: homogenous porphyritic lava. Minerals in the 2683m-core: feldspar, plagioclase, biotite, quartz (also in veins), pyrite, pyrrhotite, titanite, fluorite, apatite, adularia, epidote and garnet.

2684 m-core: massive fine grained rock of a subvolcanic nature. Minerals in the 2684 m-core: feldspar, pyroxene, biotite, magnetite, pyrite, pyrrhotite, actinolite, quartz and garnet.

2862 m-core: homogenous rock; recrystallized. Minerals in the 2862 m-core: quartz, adularia, albite, pyrite, pyrrhotite, epidote, biotite, apatite, diopside, hematite and chlorite.

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
