# Peer review of "Hydrothermal Alteration at the San Vito Area of the Campi Flegrei Geothermal System in Italy: Mineral Review and Geochemical Modeling"

_minerals, doi:10.3390/min11080810_

Round 1

Reviewer 1 Report

Dear Authors

The following text is addressed to the Authors of the manuscript: Hydrothermal alteration at the San Vito area of the Campi Flegrei geothermal reservoir in Italy: mineral review and geochemical modeling and is complemented by the important suggestions in the PDF file.

The manuscript is very interesting and presents an extremely relevant unpublished dataset from the Campi Flegrei geothermal region. An important mineralogical and geochemical study presenting the hydrothermal fluid system and also the evolution of the geothermal system in this scenario - that could be dangerous in the future!

However, we realize that the work presents a lot of data: a review of the previously published and unpublished dataset of the San Vito region, petrography, mineral chemistry, and geochemical modeling data. This set of many information is concatenated, at the end of the paper, with the numerical model.

The structure of the manuscript and the presentation of all this data seems to us somewhat undefined.

All data have quality and aspect of scientific rigor, however, we suggest that the manuscript should be divided into at least two distinct publications: i) geological characterization, lithology, mineralogy, petrography of the SAN VITO well region and ii) a small paper focused on the geochemical model based on the reference of the first work of the SAN VITO region.

               --> Another option: consider accepting the data and structure (partly) of this manuscript in its present form but in the face of a reorganization of the geological and lithological settings, mineralogy, and petrography... in the introductory section of the manuscript.

Author Response

Dear Reviewer 1,

Thank you for your comments and recommendations made to our manuscript. Below you will find responses herewith answers and modifications (written in blue).

We have basically accepted all the recommendations and we are confident that the new manuscript can satisfy your request.

Regards,

Monica Piochi and co-authors

Dear Authors

The following text is addressed to the Authors of the manuscript: Hydrothermal alteration at the San Vito area of the Campi Flegrei geothermal reservoir in Italy: mineral review and geochemical modeling and is complemented by the important suggestions in the PDF file.

The manuscript is very interesting and presents an extremely relevant unpublished dataset from the Campi Flegrei geothermal region. An important mineralogical and geochemical study presenting the hydrothermal fluid system and also the evolution of the geothermal system in this scenario - that could be dangerous in the future!

However, we realize that the work presents a lot of data: a review of the previously published and unpublished dataset of the San Vito region, petrography, mineral chemistry, and geochemical modeling data. This set of many information is concatenated, at the end of the paper, with the numerical model.

Many thanks to the Reviewer for the appreciation on our work. We hope that our work will be useful to the scientific community interested in understanding the geothermal and volcanic functioning of this dormant volcano.

The structure of the manuscript and the presentation of all this data seems to us somewhat undefined.

All data have quality and aspect of scientific rigor, however, we suggest that the manuscript should be divided into at least two distinct publications: i) geological characterization, lithology, mineralogy, petrography of the SAN VITO well region and ii) a small paper focused on the geochemical model based on the reference of the first work of the SAN VITO region.

Another option: consider accepting the data and structure (partly) of this manuscript in its present form but in the face of a reorganization of the geological and lithological settings, mineralogy, and petrography... in the introductory section of the manuscript.

Although two manuscripts should have been more productive for us, we think that merging new and existing data with modelling is the best way to provide a reviewed picture and analysis. However, we agree that the manuscript’s structuring can be improved and we have rearranged sections 1, 2, 3 (by inserting authors/references in AGIP’s report) following your suggestion.

Attachment:Reviewer 1 marked some questions on the paper.

You will find responses in the pdf, below and in the edited files in track mode.

It is certainly a very interesting and important project for the evolution of the geological knowledge of Campi Flegrei and Italia but we suggest a more direct and objective way of writing to the big picture, methods, results and the most relevant conclusions of the manuscript.

We have edited the sentence. The new sentence is “This paper focuses on fluid-rock interaction effects at the Campi Flegrei and presents relevant information requested for reactive transport simulations. In particular, we provide: 1)…”

We suggest considering a review of this last part of the session (1. Introduction). Much of what is presented here is considered a strategy approach and sampling. The strategy can be well developed in an appropriate section under MATERIALS AND METHODS. In lines 68 to 70 it is presented: "the objective of this study is to collect mineralogical information and calibrate the thermodynamic dataset with actual data to be used as input for numerical models".

Improve this part of the text.

We have followed the suggestion.

The text at lines 71-103 has been moved under Materials and Methods. We insert information on the manuscript structure at the end of the Introduction. Here we summarize the specific information and results of our study. Please, find the edit in the track file.

Figure 1

All the raised issues (cartographic references, scale, caption, coordinates) have been fixed. We have also added the location of Campi Flegrei in the Italy panel.

I didn't understand this sentence.

The sentence at line 116 has been edited and changed into: “Type and abundance of minerals that AGIP indicated to be determined by X-ray diffraction and electron microprobe analyses during drilling, were manually recovered from charts in graph papers attached to the report.”

My suggestion is to arrange the methodology in order of detection limit to promote a better understanding of the methodological sequence in the manuscript.      

Exemple: Starting with lithological descriptions and spatial relationships of lithologies (geological description) and consequently petrography, mineral chemistry and geochemistry. These data will culminate in the design of the numerical model.

We rearrange the text from introduction in the materials and methods. We think that now the section accomplishes the methodological sequence. The tracked file shows the changes made.

Assuming here that this data is your RESULTS session!

If you  no have surface data, you do not need to do this session.

This is a review and it is among our objectives of this study. We need of that and this derives from an on-going review work. We clarify this point by changing the section title “Reviewed subsurface data”. We consider that this can satisfy the request and the editorial journal policy.

Where is figure 2 D?

Fixed. It is 2b.

We suggest that Sessions 3 (including 3.1; 3.1.1 and 3.1.2) be reworded. The geological framework as well as the spatial distribution of lithologies and minerals in the SAN VITO well should be very well presented in this manuscript. The numerical geochemical model which is the fundamental objective and the major scientific contribution of this manuscript must be based on clear data (well-defined geological framework, mineralogy and petrography).

The data set presented in Session 3 can be synthesised in 1 or 2 artboards containing the well structure and the geological and mineral data. 

The sessions 3 contain the useful review arguments. It is the reason for which we wrote “data” instead of results. The treatment of data is depicted on the appearance/disappearance of alteration minerals that are then used to constrain the modeling. A description of minerals along the stratigraphic sequence would have been repetitive; it is shown in figure 2b. Note that the sequence of lithology and their mineralizations can be found in section 3 and Appendix B. However, in order to accomplish the request, we have worked in by adding a lot of information on cores (that are from our inspection of cores in the main text and Appendix C) and some considerations. Please find our editing in the tracked manuscript. The figure 2 is the synthesis of our review and dataset is in the Table S1 as excel.

This session is based on many references that should be compiled for a better understanding of this manuscript.

We have inserted a new Appendix, Appendix A, that lists references not available in literature but included in the report of AGIP. Most of text is in Italian and was as presentation in internal meetings.

Figure 2

              This figure represents an important contribution to the manuscript interpretations and presents a geological contextualization dataset summary and rocks and minerals distribution. This result is extremely important for the support of the geological framework and consequently the numerical model support.

              We strongly suggest that this figure be fixed. One way would be to improve the macroscopic images quality and consequently the macro interpretation (an example: cutting and polishing the drill cores and make a scanning high-resolution images).

              We suggest: Removing the excess of textual information and insert in the figure captions.

The figure has been rearranged. The excess of text information is now in a new Appendix C (the caption would be too long) and synthesized in caption. The macroscopic images have been enlarged. Unfortunately, we haven’t a scanner to gain the thin section images under the cross polarized and transmitted lights. Also, we do not want to cut the cores because they are residual part of our collection and cannot be damaged or reduced in size; they are too precious due to the unfeasibility of new sampling. We hope this is not a problem also in consideration of the number of images under the optical and electron microscope in figures 3 to 6 showing the textural and chemical features of studied rocks. Note, we have furnished additional description in section 3.1.

Suggestion for petrography and mineral chemistry session (optical and EDS):

Lines 287 to 305 present data that refer to figures 3, 5, Supp. 2, 4, 5, supp. 2, 5, 3, 6, 5, 6...in that order.

Arrange the text in the order of lithologies or petrographic features and organize the text into subtopics AND LIST IN THE FIGURES 3, 4, 5 ... (A, B, C, D...).

We follow your suggestion. The panels in figures 3 to 6 are now arranged in function of cores depth and recalled following their appearance in figures. You will find edits made to accomplish your suggestion in the tracked manuscript file.

Exemple: "The petrography data show..."

We follow your suggestion. Please find the edited text accordingly in the tracked file.

Structure the figure 3 and 4 (or the text) in the order of the writing (a, b, c, d ...) and at the end present the rocks that are not in the borehole. This is important information to support the model. Example: line 278

The arrangement of panels in figures 3 and 4 and their call in the text now matches. Please, find the new version in the tracked manuscript file.

Suggestion: Remove the repeated images (CP and PP) from figures 3 and 4. The extra pictures can be inserted in supplementary material.

Done. The reviewed manuscript includes the Figure S1.

The example in FIGURE 4 C and D shows sulphide in both images, and only that in D.  

line 314 refers to both images 4c and 4d

CC = CALCITE ?

Correct. We have inserted in the caption

Figures 3 to 6

We have checked the figure references throughout the text.

2 and  (7)  ?

“7” erased

Reviewer 2 Report

This Ms is an excellent piece of work analysing from the thermodynamic point of view the hydrothermal alteration in Campi Flegrei geothermal systems. Authors apply some thermodynamic softwares based on real mineralogy obtained from one drill core. Probably the extension of the Ms is to long and some information concerning petrography and stratigraphic description could be included as annexes. (this is only one suggestion).

In my opinion, the Ms highlight the main topics needed for any thermodynamic (even kynetic) modelling in geothermal research. Authors discuss all parameters neede to be into account for a thermodynamic and kynetic modellling on an active gothermal system. Conclusions are completely supported by data and results

I have no relevant comments to this Ms. Only few suggestions and one question.

I suggest to modify in the title the word "Reservoir" by "System". As authors even affirm in line 641-642. the simulations obtained in this work have a large application to the Campi Flegrei volcanic field. In consequence, I suggest to include in the title "system" rather than "reservoir" because the results obtained could be ussefull for all this geothermal system and not only for the reservoir conditions.

My question is in relation with the sentence in line 632-633. Authors affirm that "the abundance of chlorite is favoured by the K-richness of initial rock compositions". Why a K-rich system could favoured Chl formation?. Chl don't accepts K in its structure, then I don't see the correlation between K-rich lithologies and Chlorite. Please, discuss this point.

Minor comments: 

Line 95: change Authors by authors

Figure 1: where is the geothermal reservoir placed on this figure?. At least in my version I can't identify the gray color selected on the legend for the reservoir

Figure 3. please, increse its size

Figure 6: please increase its size

Line 364: change "photolith" by "protholith" (I suposses)

Line 374: change "soldalite" by "sodalite" (I syuposses)

Line 381: In figure 7 there isn't any stratigraphic reconstruction

References: please, complete reference 89.

Author Response

Dear Reviewer, below you will find our response (written in blue) to your specific comments (written in black). The manuscript has been edited accordingly.

Regards

Monica Piochi and co-authors

This Ms is an excellent piece of work analysing from the thermodynamic point of view the hydrothermal alteration in Campi Flegrei geothermal systems. Authors apply some thermodynamic softwares based on real mineralogy obtained from one drill core. Probably the extension of the Ms is to long and some information concerning petrography and stratigraphic description could be included as annexes. (this is only one suggestion).

In my opinion, the Ms highlight the main topics needed for any thermodynamic (even kynetic) modelling in geothermal research. Authors discuss all parameters neede to be into account for a thermodynamic and kynetic modellling on an active gothermal system. Conclusions are completely supported by data and results

Many thanks to the Reviewer for the appreciation on our work. We are honoured by compliments.

I have no relevant comments to this Ms. Only few suggestions and one question.

You will find our responses below.

I suggest to modify in the title the word "Reservoir" by "System". As authors even affirm in line 641-642. the simulations obtained in this work have a large application to the Campi Flegrei volcanic field. In consequence, I suggest to include in the title "system" rather than "reservoir" because the results obtained could be ussefull for all this geothermal system and not only for the reservoir conditions.

Done.

My question is in relation with the sentence in line 632-633. Authors affirm that "the abundance of chlorite is favoured by the K-richness of initial rock compositions". Why a K-rich system could favoured Chl formation?. Chl don't accepts K in its structure, then I don't see the correlation between K-rich lithologies and Chlorite. Please, discuss this point.

Right. This is from an old version of manuscript. It is now fixed.

Minor comments:

Line 95: change Authors by authors

Done.

Figure 1: where is the geothermal reservoir placed on this figure?. At least in my version I can't identify the gray color selected on the legend for the reservoir

We fix the possible problem and the shaded reddish area in legend should be now visible (at least we can see it).

Figure 3. please, increse its size

Done, also in consideration of changed suggested by reviewer 1. The figure is enlarged to maximum dimension possible in the page. The text on panel has been increased in size.

Figure 6: please increase its size

Done. The figure is enlarged to maximum dimension possible in the page. The text on panel has been increased in size.

Line 364: change "photolith" by "protholith" (I suposses)

Yes, protolith. Done.

Line 374: change "soldalite" by "sodalite" (I syuposses)

Yes, sodalite. Done.

Line 381: In figure 7 there isn't any stratigraphic reconstruction

Right. “7” erased.

References: please, complete reference 89.

Done.